# Inflammation and Wasting of Skeletal Muscles in Kras-p53-Mutant Mice with Intraepithelial Neoplasia and Pancreatic Cancer—When Does Cachexia Start?

**DOI:** 10.3390/cells11101607

**Published:** 2022-05-11

**Authors:** Wulf Hildebrandt, Jan Keck, Simon Schmich, Gabriel A. Bonaterra, Beate Wilhelm, Hans Schwarzbach, Anna Eva, Mirjam Bertoune, Emily P. Slater, Volker Fendrich, Ralf Kinscherf

**Affiliations:** 1Institute of Anatomy and Cell Biology, Department of Medical Cell Biology, Philipps-University of Marburg, Robert-Koch-Str. 8, 35032 Marburg, Germany; jan.keck@med.uni-goettingen.de (J.K.); s.schmich@gmx.net (S.S.); gabriel.bonaterra@staff.uni-marburg.de (G.A.B.); wilhelmb@staff.uni-marburg.de (B.W.); hans.schwarzbach@staff.uni-marburg.de (H.S.); annaeva@gmx.de (A.E.); mirjam.bertoune@staff.uni-marburg.de (M.B.); ralf.kinscherf@uni-marburg.de (R.K.); 2Department of General, Visceral and Pedriatic Surgery, University Clinics, Georg-August University, Robert-Koch-Str. 40, 37075 Goettingen, Germany; 3Department of Visceral, Thoracic and Vascular Surgery, University Clinics of Giessen and Marburg, Baldinger Str., 35043 Marburg, Germany; slater@staff.uni-marburg.de (E.P.S.); vfendrich@schoen-klinik.de (V.F.); 4Center for Endocrine Surgery, Schön Klinik Hamburg-Eilbek, Dehnhaide 120, 22081 Hamburg, Germany

**Keywords:** cancer cachexia, muscle atrophy, weight loss, sarcopenia, cytokines, gastrointestinal cancer

## Abstract

Skeletal muscle wasting critically impairs the survival and quality of life in patients with pancreatic ductal adenocarcinoma (PDAC). To identify the local factors initiating muscle wasting, we studied inflammation, fiber cross-sectional area (CSA), composition, amino acid metabolism and capillarization, as well as the integrity of neuromuscular junctions (NMJ, pre-/postsynaptic co-staining) and mitochondria (electron microscopy) in the hindlimb muscle of LSL-Kras^G12D/+^; LSL-TrP53^R172H/+^; Pdx1-Cre mice with intraepithelial-neoplasia (PanIN) 1-3 and PDAC, compared to wild-type mice (WT). Significant decreases in fiber CSA occurred with PDAC but not with PanIN 1-3, compared to WT: These were found in the gastrocnemius (type 2x: −20.0%) and soleus (type 2a: −21.0%, type 1: −14.2%) muscle with accentuation in the male soleus (type 2a: −24.8%, type 1: −17.4%) and female gastrocnemius muscle (−29.6%). Significantly higher densities of endomysial CD68+ and cyclooxygenase-2+ (COX2+) cells were detected in mice with PDAC, compared to WT mice. Surprisingly, CD68+ and COX2+ cell densities were also higher in mice with PanIN 1-3 in both muscles. Significant positive correlations existed between muscular and hepatic CD68+ or COX2+ cell densities. Moreover, in the gastrocnemius muscle, suppressor-of-cytokine-3 (SOCS3) expressions was upregulated >2.7-fold with PanIN 1A-3 and PDAC. The intracellular pools of proteinogenic amino acids and glutathione significantly increased with PanIN 1A-3 compared to WT. Capillarization, NMJ, and mitochondrial ultrastructure remained unchanged with PanIN or PDAC. In conclusion, the onset of fiber atrophy coincides with the manifestation of PDAC and high-grade local (and hepatic) inflammatory infiltration without compromised microcirculation, innervation or mitochondria. Surprisingly, muscular and hepatic inflammation, SOCS3 upregulation and (proteolytic) increases in free amino acids and glutathione were already detectable in mice with precancerous PanINs. Studies of initial local triggers and defense mechanisms regarding cachexia are warranted for targeted anti-inflammatory prevention.

## 1. Introduction

Cancer cachexia has been recognized as a complex, largely therapy-resistant syndrome that involves progressive muscle wasting and massively impairs the patients’ prognosis, therapy outcome, quality of life, exercise capacity and pulmonary function [1,2,3,4,5,6]. In particular, patients with pancreatic ductal adenocarcinoma (PDAC) experience a high prevalence (up to 80%) and early onset (45% at the time of diagnosis) of cachexia which may account for up to 30% of the mortality for this 4th–5th leading cause of cancer-related death [6,7,8]. Cancer-related muscle wasting has been attributed to circulating, i.e., systemic inflammatory, prooxidative, or other catabolic factors of tumor or multiple host origins [9,10,11,12,13], such as TNF, IL6, PIF, ZAG, or the more recently identified IL8 [14] or MyD88 [15]. These factors may variably trigger muscular proteasomal or other proteolytic pathway activation, inhibit protein synthesis and muscle fiber regeneration, or cause apoptosis. This may involve the activation of the NF-κB-, STAT3- or myostatin-related pathways, inhibition of the insulin-Akt pathway, upregulation of E3-ligases or autophagic genes, mitochondrial dysfunction, or other mechanisms [6,9,16,17,18,19,20,21]. Besides, inflammation-driven increases in hepatic amino acid consumption and nitrogen losses as well as high resting fat and liver energy expenditure contribute to cachexia as a catabolic multi-organ syndrome [11,12,13].

In the study of systemic inflammation [22], little is known about the dynamics of the local inflammatory microenvironment of skeletal muscles that may trigger fiber-type specific atrophy in addition to fiber type-transition, apoptosis and impaired regeneration [6]. Preliminary clinical data show a myositis-like phenotype of the rectus abdominis muscle with endomysial fibrosis and CD68+ macrophages infiltration in cachectic, compared to non-cachectic PDAC patients [23]. In addition, there is a negative correlation between CD163+ macrophage infiltration and the muscle fiber cross-sectional area (CSA) [24]. Infiltrating macrophages yield important signals for repair, satellite cell activation and myogenesis as well as angiogenesis and matrix reorganization [25]. However, macrophages may also release cytokines such as IL6 and IL-1α to activate the STAT3 signaling pathway and contribute to muscle wasting [24,26]. Cachexia-relevant myocellular responses to STAT3 activation include the upregulation of SOCS3 that promotes insulin resistance, proteolysis and mitochondria dysfunction, all of which are implicated in cancer cachexia [27,28] and its gender-related variability [29]. Indeed, the inhibition of STAT3 and SOCS3 has been shown to reverse cancer-related muscle wasting [26,30,31].

At present, it is unknown whether muscular inflammation or the expression of cytokines and SOCS3 precedes or follows the metabolic or morphological onset of muscle wasting. Moreover, it is unclear whether systemic inflammation is strictly dependent on the presence of invasive PDAC or if it may already develop in association with the inflammatory pancreatic microenvironment of precancerous intraepithelial neoplasia (PanIN). In fact, recruited M1 macrophages appear to promote not only the reversible acinar-to-ductal metaplasia (ADM), but also Kras activation-related pre-neoplastic lesions of PanINs, as well as further IL13-dependent steps towards PDAC development [32,33]. Of interest, in a preliminary study, we found significant increases in CD68+ macrophages and COX2+ cells in the liver (which receives portal venous blood from the pancreas) of mice with precursor lesions PanIN 1A-B and PanIN 2–3.

Our goal was to investigate the exact onset of local inflammatory triggers and the manifestation of skeletal muscle wasting by use of a mouse model. The established pancreas-targeted (knock-in) syngeneic Kras^G12D/+^; P53^R172H/+^; Pdx1-Cre^+/+^ (KPC) mouse model of PDAC [34,35] recapitulates mutant-based carcinogenesis with the stepwise development of PanIN and PDAC. The 2x-transgenic Kras-activating mutant mice develop mostly tumor-free PanIN stages, while transition to PDAC is slow and rare (6%). In contrast, the 3x-transgenic Kras-activating and p53-inactivating mutant mice develop PDAC at a high rate (97%) and accurately reflect its clinical profile of cachexia, metastases and hemorrhagic ascites, whereby the unrealistic tumor mass or inoculation sites (that are required for other cancer cachexia models) can be avoided. Importantly, the orthotopic inoculation of a small number of PDAC-cells from this KPC-cell line in the pancreas instantly promotes massive skeletal muscle wasting, including E3-ligase upregulation and hepatic inflammation in C57BL/6 mice [36]. Thus, the presence of PDAC is clearly the cause of inflammation and muscle wasting, as was also reported for other tumor entities [37]. However, the stepwise progression of precancerous PanIN and their transition to manifest PDAC have not yet been monitored for local cachexia triggers at the skeletal muscle tissue level.

Therefore, the present study used the KPC mouse model [34,35] to comprehensively screen the precancerous PanIN stages 1A-B and PanIN 2-3 as well as manifest PDAC with regards to fiber-type specific CSA, composition and capillarization, endo-/perimysial CD68+ macrophages and COX2+ cells, expression of pro-atrophic, -inflammatory, and pro-/anti-apoptotic, as well as angiogenic signals. Moreover, the measurements comprised intramyocellular levels of free proteinogenic amino acids, glutathione (GSH) and carnosine, as well as subgroup analyses of neuromuscular junction (NMJ) and mitochondrial ultrastructure integrity, all of which are understudied cachexia factors [6,16,38,39,40]. Our detailed muscle- and gender-specific cachexia analysis of PanIN or PDAC in KPC mice suggests that fiber atrophy is strictly PDAC–dependent and likely occurs in the absence of NMJ or mitochondrial damage. However, local inflammatory macrophage infiltration and SOCS3 upregulation are already detectable in early PanIN states and are associated with antioxidative (defense) responses warranting further studies.

## 2. Materials and Methods

### 2.1. KPC Mouse Model

Transgenic Lox-Stop-Lox (LSL)-Kras^G12D/+^ or LSL-Kras^G12D/+^ LSL-P53^R172H/+^ mice strains were interbred with Pdx1-Cre^+/+^ at the Biomedical Research Centre of the University of Marburg to obtain 2x (KC)- or 3x-transgenic (KPC) PanIN 1-3 or 3x-transgenic PDAC phenotypes on a mixed 129/SvJae/C57Bl/6 background as described by Hingurani et al. [34,35]. Wild type (WT) C57Bl/6 mice served as controls. The mice were kept at 21 (±1) °C room temperature and 55 (±5)% humidity in cages for a maximum of 5 female or male animals and provided an enriched environment, as well as alternating light/dark periods lasting 12 h. Food and water were offered ad libitum. The study was approved (MR 20/11-Nr.70/2009) by the Regional Commission of Giessen and conducted according to the regulation for animal experiments of the Philipps-University of Marburg.

Between the age of 15 and 47 weeks, mice were sacrificed by cervical dislocation and the right and left triceps surae and quadriceps muscles as well as the liver were carefully removed, shock-frozen in liquid nitrogen-cooled isopentane and stored at −80 °C. For histomorphometry and immunolocalization, serial cross-sections of 7 µm were obtained at −20 °C by a cryostat microtome (Hyrax C60, Carl Zeiss AG, Oberkochen, Germany) from the right triceps surae (gastrocnemius and soleus muscles) and quadriceps muscles as well as from the liver. Serial cryo-cross-sections from the triceps surae muscle in total were obtained to allow for the simultaneous staining of the soleus, gastrocnemius (and plantaris) muscles on the same transversal level around a maximal circumference. For transmission electron microscopy (TEM) of the mitochondrial ultrastructure, small pieces (ca. 1 mm^3^) of right gastrocnemius and soleus muscles were immediately fixed with 100% ITO-solution (2.5% glutaraldehyde, 1.25% paraformaldehyde (PFA) and 2 mM picric acid in cacodylate buffer 0.1 M, pH 7.0). Additionally, samples of the left gastrocnemius and soleus muscles were shock-frozen for measurements of the myocellular gene expression, intramyocellular amino acid and GSH level. NMJ integrity was evaluated in serial cross-sections of the right quadriceps muscle.

### 2.2. Pancreatic Histopathological Phenotype/Group Assignment

The histopathological diagnosis of PanIN stages 1A-3 as well PDAC were assessed according to [41] by two independent experienced gastrointestinal pathologists (blinded to the experimental groups). They evaluated >10 serial 4 µm cross-sections in 6 locations of the pancreatic tissue (10% formalin-fixed, paraffin-embedded; stained with hematoxylin and eosin). The mice were assigned to groups with PanIN 1A-B, PanIN 2-3 or PDAC for separate comparison to WT mice, which underwent pancreatic phenotyping as well. No tumor staging with regards to the evaluation of metastases, ascites or other features of PDAC was obtained.

### 2.3. Genotyping of the KPC Mice

Genotyping of tail tip (0.5 cm) tissue at the age of 4 weeks followed an established PCR protocol using the DNeasy Tissue Kit (Qiagen, Hilden, Germany) for extraction of the genomic DNA and the following set of forward (FW) and reverse (REV) primers (sequence 5′→3′): K-ras WT FW (GTC GAC AAG CTC ATG CGG GTG), K-ras 006 WT REV (CCT TTA CAA GCG CAC GCA GAC TGT AGA), K-ras Mutant FW (AGC TAG CCA CCA TGG CTT GAG TAA GTC TGC A), K-ras 006 Mutant REV (CCT TTA CAA GCG CAC GCA GAC TGT AGA), P53 WT FW (TTA CAC ATC CAG CCT CTG TGG), P53 WT REV (CTT GGA GAC ATA GCC ACA CTG), P53 Mutant FW (AGC TAG CCA CCA TGG CTT GAG TAA GTC TGC A), P53 Mutant REV (CTT GGA GAC ATA GCC ACA CTG), Cre FW (CCT GGA AAA TGC TTG TGT CCG) andCre REV (CAG GGT GTT ATA AGC AAT CCC).

### 2.4. Fiber Size, Composition and Capillarization

Histochemical fiber-typing was based on acid-sensitive myofibrillar ATPase staining (Adenosine triphosphatase, Sigma-Aldrich Co. LLC, St. Louis, MO, USA) after preincubation at pH 4.6 (5 min, room temperature), enabling the identification of type 2, type 2a or type 2x (including 2b) fibers. To determine fiber type-specific capillary contacts beside capillary density and capillary to fiber ratio an adjacent transverse section was stained for capillaries using horseradish peroxidase (HRP)-conjugated Isolectin B4 of Bandeiraea simplicifolia (BSI-B4; Sigma-Aldrich Co. LLC) with 3,3′-diaminobenzidine (DAB, Sigma-Aldrich Co. LLC) detection and additional counterstaining of the nuclei with Mayer’s hematoxylin (Carl Roth GmbH & Co. KG, Karlsruhe, Germany).

The fiber CSA of type 2, 2a and 2x/b were assessed together with fiber composition in terms of fractional counts (%) and fractional area (%), as well as the fraction of the endomysial (interstitial) area in relation to the muscle tissue area in total. The analysis of capillarization was performed in a BSI- B4-stained section, enabling the identification of fiber type from the directly neighboring ATPase-stained section, and included capillary density (CD, capillaries per mm^2^) and the capillary contacts of each single fiber type 2, 2a or 2x/b (including shared capillaries), as well as the number of fiber-adjacent capillary contacts per unit fiber CSA representing an index of nutritive capillary supply. The mean fiber counts analyzed per rectangle were similar between the groups WT, PanIN 1A-B, PanIN 2-3 and PDAC, amounting to 137.0 ± 7.4, 157.3 ± 7.6, 145.0 ± 7.4, and 146.9 ± 6.6 in the gastrocnemius as well as 237.4 ± 14.0, 223.9 ± 12.7, 238.6 ± 22.8, and 247.6 ± 13.2 in the soleus muscle, respectively. Notably, in the gastrocnemius muscle, the analysis was limited to its larger homogenous ‘white‘ (i.e., superficial) area that consists of type 2x/b fibers only.

### 2.5. Immunohistochemistry of CD68+ Macrophages and COX2+ Cells

For the immunolocalization of CD68+ macrophages, 7 µm cryo-cross-sections of skeletal muscle and liver were fixed with 4% PFA/phosphate buffered solution (PBS; 10 min, ambient temperature), blocked with 1% normal swine serum (Dako Deutschland GmbH, Hamburg, Germany) in PBS and incubated with monoclonal CD68 primary antibody (rat anti-mouse, 1:50; AbD Serotec, Kidlington, UK). CD68+ cells were detected by polyclonal HRP-conjugated anti-rat antibodies (goat anti-rat, 1:100; AbD Serotec) and incubated with DAB solution (Sigma-Aldrich Co. LLC) after the suppression of endogenous peroxidase activity with 3% H_2_O_2_ in PBS. Nuclei were counterstained with Mayer’s hematoxylin. For visualization of COX2+ cells, 7 µm cryo-cross-sections were processed as described above for CD68+ cells and incubated with polyclonal COX2 primary antibodies (rat anti-mouse, 1:50; Abcam, plc, Cambridge, UK). The detection of COX2+ cells was performed using HRP-conjugated antibodies (goat anti-rat, 1:100; AbD Serotec) and incubation with DAB solution. The endo-/perimysial CD68+ macrophage and COX2+ cell infiltrations were determined as cell count per count of muscle fibers within a rectangle area containing >100 muscle fibers. The CD68+ macrophage and COX2+ cell densities around 4 central veins in the liver were measured as cell count per circle area, defined by its radius of three times the central vein diameter. Only the areas containing intact muscle or liver tissue were analyzed by use of the Fiji software (National Institute of Health, Bethesda, MD, USA) and applied to microscopic images taken at a 200-fold magnification with Zeiss Axio Imager.M2 microscope (Carl Zeiss AG).

### 2.6. Density and Integrity (Pre-/Post-Synaptic Co-Staining) of Neuromuscular Junctions (NMJ)

To assess the density of NMJs in the quadriceps muscle, every 7th out of 70 serial cross-sections was treated for 10 min with 3% H_2_O_2_ to block endogenous peroxidases and, thereafter, NMJ postsynapses stained by incubation with biotin-XX-conjugated of alpha-bungarotoxin (α-BTX) and, subsequently, HRP-conjugated Streptavidin (Jackson ImmunoResearch Laboratories. Inc., West Grove, PA, USA) in the presence of the substrate DAB. Nuclei were counterstained with Mayer’s hematoxylin. NMJ count per area was determined in digital images (200-fold magnification) obtained by the Zeiss Axio Imager.M2 microscope combined with Axio-Cam HRc/AxioVision (Carl Zeiss GmbH), covering 115.0 ± 23.3, 218.4 ± 40.8, 235.0 ± 71.6, and 269.0 ± 80.8 NMJs per mouse in the groups WT, PanIN 1A-B, PanIN 2-3, and PDAC, respectively.

To evaluate the integrity of NMJs, i.e., to screen for possible denervation states in subgroups of mice (WT: *n* = 5, PanIN 1A-B: *n* = 5, PanIN 2-3: *n* = 5, and PDAC: *n* = 4), immunofluorescent double staining was carried out in every 6th of 60 serial 7µm transverse sections of the vastus muscle with a biotin-XX-conjugated α-BTX (Thermo Fischer Scientific, Schwerte, Germany) and vesicular acetylcholine transporter (vAChT) rabbit antibody (Lee Eiden, Lot-No.: bleed 6/97). Briefly, the sections were fixed (10 min 4% PFA/PBS), blocked (30 min 1% BSA/PBS) and incubated overnight at 4 °C with biotinylated α-BTX (1:500) or vAChT antibody (1:1000) in a humidified chamber, and thereafter washed and incubated for 30 min with Cy3-conjugated streptavidin (1:200, Dianova GmbH, Hamburg, Germany) or Alexa Fluor^®^ 488 labeled donkey anti-rabbit IgG (1:200, MoBiTec GmbH, Goettingen, Germany), respectively. Non-specific staining was controlled by similar handling without the primary antibody or histochemical staining. Confocal images were obtained using the scanning laser microscope C2 (Nikon GmbH, Düsseldorf, Germany) based on the software NIS-Elements AR 4.30.01 (Laboratory Imaging). NMJs were scanned at 630-fold magnification and 250 Hz with an image size of 1024 × 1024 pixels, covering 222.4 ± 69.9, 305.8 ± 42.2, 305.2 ± 93.9, and 235.0 ± 71.6 double-stained NMJs per mouse in the groups WT, PanIN 1A-B, PanIN 2-3, and PDAC, respectively. The histomorphometrical determination of α-BTX- and vAChT- immunolabeled areas and their absolute or percentage (% of α-BTX area) overlap was carried out by means of Fiji software.

### 2.7. Transmission Electron Microscopy (TEM)

The samples that were fixed in 100% ITO-solution were post-fixed in osmium tetroxide (1%), dehydrated afterwards with ethanol/propylene oxide, and embedded in EPON™ 812 (SERVA GmbH, Heidelberg, Germany). Ultrathin sections (60–80 nm) were cut using a Reichert Ultracut S ultramicrotome (Leica Microsystems, Wetzlar, Germany) and contrasted with 4% uranyl acetate and lead citrate. The ultrastructural integrity of the muscle mitochondria was observed with a TEM Zeiss EM 10 C, Carl Zeiss GmbH, Jena, Germany). Pictures were taken with an Image ASP System (SYSPROG, Minsk, Belarus), as previously described [42]. Using ImageJ software (Scion Image, National Institutes of Health, Bethesda, MD, USA), ten images per mouse were randomly taken at a magnification of 25,000×. Images were analyzed with regards to alterations in the mitochondrial ultrastructure that were categorized as “normal” or “damaged” as follows: Mitochondria with a >50% loss of the cristae, and/or those with >50% disruption of the outer membrane (OMM) were assigned to “damaged” or otherwise to “normal”. Results per mouse were presented as the ‘damaged’ fraction (%) of the total mitochondria count.

### 2.8. Intramyocellular Levels of Reduced or Oxidized Glutathione (GSH, GSSG) and of Free Amino Acids

Homogenized samples (20–30 mg) from the gastrocnemius muscle were deproteinized with 500 µL 2.5% sulfosalicylic acid (SSA), sonificated and centrifugated for 10 min at 5 °C. The supernatant was used to determine the intramyocellular levels of the acid-soluble amino acids using a high-performance liquid chromatography (HPLC; Biochrom 30plus, Onken, Gründau, Germany) technique with the ninhydrin colorimetric detection method, as described earlier [43]. Furthermore, the content of the total and reduced GSH and its oxidized disulfide GSSG were determined according to Tietze’s assay [44] as previously described [43]. Reduced GSH was calculated by the subtraction of GSSG from the total GSH. The total protein content of the pellet corresponding to the supernatant volume was quantified by the colorimetric Bio-Rad BCA (bicinchoninic acid) protein assay (Biorad Laboratories, Munich, Germany) according to manufacturer’s instructions in order to normalize the intracellular content of GSH, GSSG and amino acids.

### 2.9. Real-Time Quantitative Transcription Polymerase Chain Reaction (RT-qPCR)

For RT-qPCR, the gastrocnemius and soleus muscle samples were processed as recently described [42]. Briefly, a peqGOLD Isolation Systems TriFast™ (PEQLAB Biotechnologie GmbH, Erlangen, Germany) was used for RNA extraction according to the manufacturer’s protocol. RNA concentration was determined using a NanoDrop 2000c (Thermo Fisher Scientific Inc., Waltham, MA, USA) by optical density (OD_260nm_). RNA integrity was confirmed using an RNA 6000 NanoChip kit on an Agilent 2100 Bioanalyzer (Agilent Technologies, Waldbronn, Germany). An aliquot of total RNA (1 µg) was treated (30 min; 37 °C) with one unit DNAse (Thermo Fisher Scientific Inc.). The treated RNA was employed to perform the RT-qPCR using oligo primer (dT)_12–18_ (Agilent Technologies, Waldbronn, Germany); 20 units of the Affinity Script multiple temperature cDNA synthesis kit (Agilent Technologies Inc.); 24 units of the Ribo Lock™ RNAse inhibitor (Thermo Fisher Scientific Inc.); and 4 mM of the dNTP-Mix (Agilent Technologies Inc.). The reverse transcription was performed at 42 °C for 1 h. The cDNA was used for qRT-PCR using the QuantiTect/primer Assays (Qiagen N.V., Venlo, The Netherlands). Duplicates of cDNAs samples were amplified with the Agilent™ Brilliant III Ultra-Fast SYBR^®^ Green QPCR Master-Mix (Agilent Technologies Inc.). The thermal profile consisted of 1 cycle of 3 min. at 95 °C, followed by 45 cycles at 95 °C (10 s) and 60 °C (20 s). The relative amount of each sample was calculated from the respective standard curves based on pooled cDNA using the Mx3005P™ QPCR System (Agilent Technologies Inc.). Specificity of the amplified product was confirmed by melting curve analysis (55–95 °C).

The presently used primers were purchased from Qiagen N.V. (Venlo, The Netherlands) and comprised: Actin Beta (ACTB, 77 bp, QT01136772); retention in endoplasmatic reticulum sorting receptor-1 (RER1, 86 bp, QT00146580); Atrogin-1/F-Box Protein 32 (FBXO32, 103 bp, QT00158543); muscle RING finger-1 (MuRF1/TrIM63, 116, QT00291991); B-cell lymphoma 2 Apoptosis Regulator (BCL-2, bp104, QT02392292); BCL-2 Associated X (BAX, 76 bp, QT00102536); caspase-3 (Casp3, 150 bp, QT01164779); CD68-molecule (CD68, 67 bp, QT00254051); cyclooxigenase-2 (COX2/PTGS, 95 bp, QT00165347); interleukin-1β (IL-1β, 150 bp, QT01048355); interleukine-6 (IL-6, 128 bp, QT00098875); tumor necrosis factors-α (TNFα, 112 bp, QT00104006); suppressor of cytokine signaling-3 (SOCS3, 90 bp, QT02488990); peroxisome proliferator-activated receptor gamma coactivatror-1α (Ppargc1α/PGC-1α, 63 bp, QT00156303); matrix metoallopeptidase-9 (MMP-9, 84 bp QT00108815); monoamine oxidase-A (MAO-A, 81 bp, QT00109326); monoamine oxidase-B (MAO-B, 93 bp, QT00145124); myogenin (MyoG 115 bp, QT00112378); paired box 7 (PAX7, bp 135, QT00147728); and vascular endothelial growth factor-A (VEGF-A, 117 bp, QT00160769).

Gene expressions were normalized for most stable housekeeper ACTB for the gastrocnemius and RER1 for soleus muscle (identified by the NormFinder software, http://moma.dk/normfinder-software, accessed on 10 March 2022) and expressed relative to WT.

### 2.10. Statistics

Data are presented as mean ± standard error of the mean (SEM). To detect a significant and independent impact of PDAC or tumor-free PanIN stages 1A-3 in total as compared to WT, controlling for the factors gender (male or female) and factor age (< or >20 weeks), a univariate ANOVA was applied. Differences between PDAC or all tumor-free PanIN stages 1A-3 or 2-3 only and WT, or between PDAC and PanIN stages were (posthoc) detected by Student’s t-test or, in case of no normal distribution, by the Mann–Whitney U test. Significant bivariate linear correlations between individual measures were presented as scatterplots with an indication of the groups’ assignments, regression line, Pearson’s correlation coefficient r, and the *p* values. Multiple regression was applied to assess the independent impact of (inflammatory) pancreatic (histological) phenotype (WT = 0, PanIN 1A-B = 1, PanIN 2-3 = 2 and PDAC = 3) and of the factor age (weeks) on the CD68+ and COX2+ cell infiltration of the gastrocnemius and soleus muscles. A *p* ≤ 0.05 was considered as statistically significant. Given the explorative nature of the study, no Bonferroni correction was intended. All statistical procedures were performed by SPSS (version 27, IBM Munich, Germany).

## 3. Results

### 3.1. Study Population

As shown in Table 1, data were grouped according to the pancreatic histological phenotype, i.e., WT (normal pancreatic phenotype, i.e., no intraepithelial neoplasia or invasive PDAC), PanIN 1A-B, PanIN 2-3 or invasive PDAC. Mice presenting with PanIN 1A-B were predominantly, but not exclusively, 2x-trangenic (Kras-PDX) mice. PanIN 2-3 lesions were found in both, 2x-transgenic and 3x-trangenic (Kras-p53-PDX) mice, and invasive PDAC occurred in 3x-transgenic mice only, as expected. Body weight on the day of sacrifice was not significantly impacted by PanIN or PDAC compared to WT. Because female mice had significantly lower body weight than male mice and age was significantly higher in mice with PDAC or PanINs compared to WT (Table 1), we analyzed the effect of PDAC or PanINs on body weight with control for age and sex by ANOVA. This procedure detected a significant effect of PDAC (*p* = 0.002) and of sex (*p* < 0.001), but not of age (*p* = 0.146) on body weight. Moreover, body weight was significantly impacted by PanIN stages (*p* = 0.002) when controlled for sex (*p* < 0.001) and age (*p* < 0.001).

### 3.2. Fibers Size (Gastrocnemius and Soleus Muscle) and Fiber Composition (Soleus Muscle)

As a main factor in muscle wasting, reductions in mean skeletal muscle fiber CSA are presented for the gastrocnemius muscle in Figure 1 and the soleus muscle in Figure 2 (Figure 2a: type 1, Figure 2b: type 2a, and Figure 2c: type 2x fibers). Compared to WT, CSA of gastrocnemius muscle type 2x fibers (Figure 1a) and of soleus muscle type 2a fibers (Figure 2b) significantly decreased with manifest PDAC compared to WT within the total study population. PDAC induced a significant reduction in fiber CSA by 29.64% in the gastrocnemius muscle of female, but not male mice (Figure 1a). In contrast, soleus muscle type 1 (Figure 2a) and type 2a (Figure 2b) fibers showed a significant CSA reduction in males only, while its 2x fibers CSA (Figure 2c) remained unchanged in males and females.

The presence of PDAC resulted in an almost significant (*p* = 0.056) decrease in fiber CSA in the gastrocnemius muscle when controlled for the factor sex (*p* = 0.238) by ANOVA. However, with additional control for age (*p* = 0.01, assigning mice to age > or <20 weeks, see ‘Section 2.10′), the pro-atrophic impact of PDAC reached significance (*p* = 0.020), thereby significantly interacting with age (*p* = 0.032) with little effect left for sex (*p* = 0.630).

In the soleus muscle, two-factorial ANOVA showed that PDAC, independently of sex (females vs. males), indeed significantly decreased CSA of fiber type 1 (Figure 2a; PDAC: *p* = 0.041, sex: *p* = 0.017) and 2a (Figure 2b; PDAC: *p* = 0.019, sex: *p* = 0.007) compared to WT. However, PDAC did not significantly impact CSA of type 2x (*p* = 0.946) which was significantly lower in females than in male mice (*p* = 0.021). With additional control for the factor age, PDAC still significantly contributed to a decrease in CSA of type 1 fibers (PDAC: *p* = 0.005, sex: *p* = 0.012, age: *p* = 0.042) and type 2a fibers (PDAC: *p* = 0.044, sex: *p* = 0.010, age: *p* = 0.565) in soleus muscle. This approach also detected a significant CSA-decreasing effect of PDAC (*p* = 0.035) on type 2x fibers, which was independent of the factor sex (*p* = 0.006) and the factor age (*p* = 0.082).

Notably, no significant atrophy was detected in both PanIN stages (without PDAC) in any of these fiber types and muscles under study, except for gastrocnemius muscle fibers with PanIN 1A-B in females (Figure 1a). Further, when applying ANOVA with control for sex and age (as described above for testing the impact of PDAC), no significant impact of PanIN stages 1A-3 or of PanIN stage 2-3 was detected on the CSA of gastrocnemius fibers or any soleus muscle fiber type.

Thus, as a main finding, PDAC, but not the tumor-free PanIN stages (in 2x- or 3x-transgenic mice), was associated with a significant reduction in mean fiber CSA compared to WT. This pro-atrophic impact of PDAC proved to be independent of the well-known decreasing effect of female vs. male sex on fiber CSA as well as of age within the presently analyzed age span.

Regarding changes in soleus muscle fiber composition (Figure 3a–c), a significant increase was observed in the rather small fraction of type 2x fibers with PDAC and PanIN 1A-B (Figure 3c) and associated with decreases in type 2a fiber fraction with PDAC, with significant differences in male mice only (Figure 3b).

### 3.3. Capillarization of Gastrocnemius and Soleus Muscles

Capillarization in terms of capillary density, -to-fiber-ratio, or fiber type specific contacts was unaffected with PDAC or with PanIN stages in both muscles under study (Table 2), irrespective of control for sex or, additionally, for age by ANOVA. Likewise, the ratio of fiber capillary contacts/CSA, which was calculated as a marker of capillary supply for each fiber type, did not significantly differ between mice with PDAC- or PanINs and WT.

### 3.4. Density of CD68+ Macrophages and COX2+ Cells in Gastrocnemius and Soleus Muscles

The density of CD68+ macrophages (count per muscle fiber count) in the gastrocnemius muscle (Figure 4a,b) was found to be significantly and progressively increased 2.8-fold, 4.0-fold, and 5.5-fold, with PanIN 1A-B, PanIN 2-3, and PDAC, respectively, as compared to WT. This proinflammatory impact of PanIN 1A-3 or of PDAC was significant with (*p* = 0.007 or *p* < 0.001, respectively) and without (*p* = 0.006 or *p* < 0.001) control for the factor sex by ANOVA.

Similarly, in the soleus muscle, CD68+ macrophage density significantly increased 2.7-fold, 3.3-fold, and 9.2-fold with PanIN 1A-B, PanIN 2-3, and PDAC, respectively (Figure 4c,d), with the proinflammatory effect of PanIN 1A-3 overall or PDAC being significant with (*p* = 0.005 or *p* < 0.001, respectively) and without (*p* = 0.005 or *p* < 0.001) consideration of the factor sex by ANOVA. Linear regression analyses, including pancreatic (histological) phenotype (defined as WT = 0, PanIN 1A-B = 1, PanIN2-3 = 2 and PDAC = 3), age (weeks) and sex (males vs. females) revealed a significant, independent effect of the pancreatic phenotype on CD68+ macrophage density in the gastrocnemius and soleus muscle (*p* < 0.001 and *p* < 0.000, respectively) with no significant impact of sex or age (not shown).

Furthermore, in the gastrocnemius muscle COX2+ cell density (count per muscle fiber count) significantly increased 1.8-fold and 1.9-fold with PanIN 1A-B and PDAC, respectively, but not with PanIN 2-3, when compared to WT (Figure 5a,b), with the impact of PanIN 1A-3 or PDAC being significant with (*p* < 0.001 or *p* < 0.001, respectively) and without (*p* < 0.001, *p* < 0.001) control for sex by ANOVA. In the soleus muscle, COX2+ cell density also showed a significant 1.6-fold and 1.7-fold increase with PanIN 1A-B, and PDAC, respectively, with no such effect observed with PanIN 2-3 (Figure 5c,d). Again, the effects of PanIN 1-3 overall or of PDAC were significant with (*p* = 0.012 or *p* = 0.005, respectively) or without (*p* = 0.012 or *p* = 0.012) consideration of the factor sex by ANOVA. Control for age (weeks) by linear regression showed that pancreatic (histological) phenotype (defined as WT = 0, PanIN 1A-B = 1, PanIN2-3 = 2 and PDAC = 3) revealed a significant age-independent effect on COX2+ macrophage density in the gastrocnemius muscle (*p* = 0.013, age: *p* = 0.578), but not in the soleus muscle (*p* = 0.143, age: *p* = 0.283) with no significant additional impact of sex.

### 3.5. Perivenous Density of CD68+ Macrophages and COX2+ Cells in the Liver

Interestingly, in the liver, CD68+ macrophage density (count per perivenous area) was found to be significantly increased 3.9-fold, 5.1-fold, and 5.2-fold with PanIN 1A-B, PanIN 2-3 and PDAC, respectively (Figure 6a,b). Multiple regression showed that pancreatic phenotype (defined as WT = 0, PanIN 1A-B = 1, PanIN 2-3 = 2 and PDAC = 3) significantly (*p* = 0.001) and independently of sex (*p* = 0.377) or age (*p* = 0.164) impacted hepatic CD68+ macrophage density. Importantly, the above-mentioned CD68+ macrophage density in the gastrocnemius muscle was significantly positively related to perivenous liver CD68+ macrophage density (r = 0.509, *p* < 0.001; Figure 6c), and a similar relation was found between CD68+ macrophage density in the soleus muscle and the liver (r = 0.346, *p* < 0.016; not shown).

Moreover, COX2+ cell density also showed a significant increase by 2.3-fold, 2.2-fold and 3.5-fold with PanIN 1A-B, PanIN 2-3 and PDAC, respectively (Figure 6d,e). According to multiple regression, the pancreatic phenotype (defined as WT = 0, PanIN 1A-B = 1, PanIN2-3 = 2 and PDAC = 3), significantly (*p* < 0.001) and independently of sex (*p* = 0.466) or age (*p* = 0.920), impacted hepatic COX2+ cell density. Furthermore, interestingly, a significant positive correlation was found of COX2+ cell density in the soleus muscle (r = 0.617, *p* < 0.001; Figure 6f), but not in the gastrocnemius muscle (r = 0.207, *p* = 0.189, not shown), to perivenous liver COX2+ cell density.

### 3.6. Gene Expression in Gastrocnemius and Soleus Muscle

Among the E3 ligases that were used as atrophy markers in the gastrocnemius muscle of PDAC mice, Atrogin-1, but not MuRF1 expression, showed an almost significant (*p* = 0.065) 2.8-fold increase compared to WT when controlled for the impact of sex. A smaller trend (2.1-fold) towards an Atrogin-1 increase was observed with PanIN 2-3 (Table 3a). In the soleus muscle (Table 3b), PDAC was associated with a 1.7-fold increase (*p* = 0.056) in MuRF1 (significantly interacting with sex by ANOVA), but not in Atrogin-1 expression. Moreover, in the gastrocnemius muscle, neither PDAC nor PanIN stages significantly affected the expression of Casp3, BAX or BCL2 (Table 3a) when compared to WT, while gender significantly affected Casp3 and BAX. In the soleus muscle, PDAC or sex showed no significant effect on the expression of Casp3, BAX or BCL2 (Table 3b), while p62 was significantly affected by sex only.

Among the pro-inflammatory signals in the gastrocnemius muscle, significant PDAC-related increases were found for CD68 (2.6-fold), TNFα (1.7-fold) and SOCS3 (2.7-fold, Figure 7) when controlled for the factor sex, while the >3-fold increases in Il-1β and COX2 expression failed to be significant by ANOVA (Table 3a). Interestingly, beside COX2, TNFα and SOCS3 expressions were also significantly increased in PanIN 1A-B (Figure 7, Table 3a), whereby only the increase in SOCS3 reached significance with combined PanIN stages and control for sex by ANOVA. In the soleus muscle, IL-1β expression showed a highly significant seven-fold increase with PDAC independently of the factor sex, while >2-fold increases by trend only were observed for SOCS3 and COX2 but no other signal under study (Table 3b). The presumable anti-inflammatory transcript of PGC1a remained unaffected by PDAC or PanIN in the gastrocnemius muscle or PDAC in the soleus muscle (Table 3a,b). In the soleus muscle, myogenic (myogenin, Pax7) or angiogenic signals (VEGFA) remained unaffected by PDAC (Table 3b), however, myogenin was significantly impacted by sex.

### 3.7. Free Proteinogenic Amino Acids in in Gastrocnemius Muscle

As shown in Table 4, the intramyocellular pool of 18 proteinogenic amino acids (without undetectable cystine and tryptophan) in the gastrocnemius muscle showed a significant increase with both, PanIN 1A-B and PanIN 2-3 compared to WT mice, which, according to ANOVA, was independent of the simultaneous significant impact of sex, whereas a considerably smaller effect of PDAC failed to be significant with or without control for sex. Table 4 also presents a subgroup of proteinogenic amino acids with protein-anabolic signaling potential. Here leucine, valine, proline and alanine, and, in part, glutamine and serine, showed significant increases in intramyocellular levels with PanIN 1A-B and/or PanIN 2-3 stages as well as with both combined PanIN stages when compared to WT levels with control for the impact of sex by ANOVA. Additionally, there were considerably smaller increases in all amino acid levels with PDAC as compared to WT levels. Only in the case of valine and, when controlled for sex, in the case of leucine and glutamine were these differences significant.

### 3.8. Antioxidants GSH and Carnosine as Well as Precursor Amino Acids in Gastrocnemius Muscle

As presented in Table 5, the intramyocellular total and reduced GSH in the gastrocnemius muscle were significantly increased 1.2-fold in mice with PanIN stages, but not with PDAC. The latter showed a significant decrease compared to PanIN 2-3, approaching control values of WT. GSH increases were associated with an out-of-proportion 2.0-fold increase in GSSG, reaching significance in PanIN 1A-B, which indicated a strong trend towards an oxidative shift in GSH redox state in terms of a decrease in the reduced GSH/GSSG ratio with PanIN (0.70-fold), and with PDAC (0.79-fold). Regarding GSH precursor amino acids in the gastrocnemius muscle, unfortunately, the thiol compound cysteine was not determined. However, its disulfide cystine was not detectable with any of the conditions WT, PanIN 1A-B, PanIN 2-3 or PDAC. The glutamate level in the gastrocnemius muscle remained largely unaffected by PanIN stages or PDAC (though significantly impacted by sex according to ANOVA), while glycine was found to be significantly 1.3-fold increased with PanIN 2-3, or overall, with PanIN stages 1A-3, but not with PDAC.

Furthermore, intramyocellular carnosine levels in the gastrocnemius muscle revealed a 1.3-fold and 1.2-fold increase with both, PDAC and PanIN 1-3, respectively, which reached significance only upon control for a highly significant impact of sex (Table 5). Histidine—as one of the two carnosine precursors—showed a significant sex-independent 1.3-fold increase in both PanIN groups, while the 1.2-fold increase with PDAC reflected a significant interaction with sex.

### 3.9. Correlations between Inflammatory Signals, SOCS3 Expression, and GSH Redox State in Gastrocnemius Muscle

SOCS3 expression was significantly positively related to CD68 expression (r = 0.352 *p* < 0.05), however, it was not significantly positively related to CD68+ macrophage infiltration in the gastrocnemius muscle (data not shown). Moreover, significant correlations existed between the expressions of Il-1β (r = 0.447 *p* < 0.01), Il-6 (r = 0.644 *p* < 0.001) or Atrogin-1 (r = 0.452 *p* = 0.001) and the expression of CD68 (data not shown). Furthermore, SOCS3 expression was significantly related to COX2+ cell density in the gastrocnemius muscle (r = 0.394 *p* < 0.05) (Figure 8a). Among the cytokines under study, IL-6 was found to be significantly related to COX2 expression (r = 0.377 *p* < 0.05) (data not shown).

### 3.10. Correlation of Free Amino Acids to Atrogin-1 Expression and of GSH to Glycine Levels in Gastrocnemius Muscle

Significant positive correlations (r > 0.3) were found between a majority of proteinogenic amino acid levels and atrogin-1 expression in the gastrocnemius muscle (asparagine: r = 0.603 *p* < 0.001; isoleucine: r = 0.587 *p* < 0.001; histidine: r = 0.581 *p* < 0.001; phenylalanine: r = 0.558 *p* < 0.001 (Figure 8c); glutamine: r = 0.539 *p* < 0.001; leucine: r = 0.494 *p* < 0.01; valine: r = 0.416 *p* < 0.01; methionine: r = 0.416 *p* < 0.01; proline: r = 0.321 *p* < 0.05; serine: r = 0.320 *p* < 0.05), while no correlation existed between muscle fiber CSA and atrogin-1 expression within this muscle. These correlations were also found when analyzing the PanIN 1A-B, PanIN 2-3 and WT only, i.e., without PDAC (asparagine: r = 0.683, *p* < 0.001; isoleucine: r = 0.538 *p* < 0.01; histidine: r = 0.586 *p* < 0.001; phenylalanine: r = 0.537 *p* < 0.001; glutamine: r = 0.620 *p* < 0.001; leucine: r = 0.427 *p* < 0.01; valine: r = 0.348 *p* = 0.051; methionine: r = 0.416 *p* < 0.01; proline: r = 0.413 *p* < 0.05; serine: r = 0.420 *p* < 0.05). Moreover, significant positive correlations existed between the total GSH and the pooled proteinogenic acids (r = 0.447 *p* < 0.01) or its precursor amino acid glycine (r = 0.473 *p* < 0.01, Figure 8d) in the gastrocnemius muscle. However, no significant positive correlations existed between GSH and glutamate.

### 3.11. Integrity of NMJ in Quadriceps Muscle

As shown in Table 6, the density of NMJs within the quadriceps muscle was not significantly altered with PanIN 1A-B, PanIN 2-3, or PDAC compared to WT mice in explorative small subgroups. Moreover, no PanIN- or PDAC-related effects were observed with the α-BTX+ area of postsynapsis, the vAChT+ area of presynapses or their overlap in absolute or in percentage terms.

### 3.12. Mitochondrial Ultrastructural Integrity

Within small explorative subgroups of WT and PDAC, the fraction of mitochondria with ultrastructural damage (according to criteria defined in the ‘Methods’ section for 25,000× TEM images) was found to be similar between PDAC and WT mice in the gastrocnemius muscle (62.21 ± 4.27%, *n* = 5 and 60.21 ± 3.68%, *n* = 6, respectively, with 230.8 ± 44.9 and 220.0 ± 50.41 mitochondria analyzed per mouse; data not shown) as well as in the soleus muscle (60.62 ± 2.41, *n* = 7 and 58.87 ± 3.11%, *n* = 5, respectively, with 311.3 ± 42.0 and 364.6 ± 25.7 mitochondria analyzed per mouse; data not shown).

## 4. Discussion

The present study used the established 3x-transgenic (KPC) and 2x-transgenic (KC) mouse model of PDAC including its precancerous PanIN stages [34,35,45] to analyze muscle wasting as the main feature of cachexia with regards to:(a)A still lacking detailed description of PDAC-related changes in fiber type-specific CSA, capillarization, NMJ integrity, amino acid and glutathione levels, mitochondrial ultrastructure, and pro-inflammatory, -apoptotic and -atrophic gene expression in fast- and slow-twitch skeletal muscles;(b)The onset of initial, likely inflammatory, changes in skeletal muscle phenotypes in relation to the progression of precancerous PanIN stages from 1A-B to 2-3 (in 2x and/or 3x-transgenic mice), and the transition to invasive PDAC (3x-transgenic mice).

Ad (a) As a main finding in the present study, a significant reduction in fiber CSA was only detectable with invasive PDAC, but not with preceding PanIN 1A-B or PanIN 2-3 stages, irrespective of the 2x or 3x-transgenic genotype. Because this was true for both, the fast-twitch gastrocnemius and the (mixed) slow-twitch soleus muscle, muscle wasting in terms of fiber atrophy can be concluded to be dependent on the presence of invasive PDAC. This was paralleled by an increased expression of MuRF1 in the soleus muscle (*p* = 0.056) and by a strong trend of Atrogin-1 in the gastrocnemius (*p* = 0.065) muscle (Table 3a,b), i.e., of (transient) surrogate markers of proteasomal protein breakdown [4,46]. These data are compatible with the activation of a proteasomal proteolysis via nuclear translocation of the FOXO1/3, the E3-ligase transcription factor, occurring in response to local IL1β (presently increased in the soleus muscle) or to the inhibition of the IGF1-Akt-signaling pathway. The latter may occur, for example, via interference with the significant local upregulation of TNFα (as observed with control for the impact of sex). As another important signal leading to E3-ligase expression likely via the inhibition of Akt, SOCS3 has been implicated in cancer cachexia [27,29,47], age-related sarcopenia as well as metabolic syndrome [48,49]. Indeed, SOCS3 was presently found to be significantly upregulated 2.7-fold in gastrocnemius muscle with PDAC compared to WT mice. As demonstrated by inhibition, IL6-dependent STAT3 signaling may involve SOCS3 upregulation in E3-ligase-dependent proteasome activation with cancer cachexia [25,30], as well as with denervation [50,51]. While the local expression of IL6 presently failed to significantly increase with PDAC, we clearly demonstrated that pro-inflammatory signals occur in both muscles: Highly significant 5.5- and 9.2-fold increases in pro-inflammatory CD68+ M1-macrophages were detected with PDAC in the gastrocnemius and soleus muscles, respectively, which was paralleled by a significant increase in CD68 transcripts (controlled for the impact of sex) in the gastrocnemius muscle. This was associated with a 1.9- and 1.7-fold enhancement of COX2+ cells in the gastrocnemius and soleus muscles, respectively, in PDAC compared to WT mice, and with a parallel trend in COX2 transcripts.

Ad (b) As an important novel finding, a significant SOCS3 upregulation in the gastrocnemius muscle as well as proinflammatory CD68+ M1-macrophages and COX2+ cell infiltrations of the gastrocnemius and soleus muscles were detected in mice with PanIN 1A-B and PanIN 2-3 stages, i.e., in the absence of significant fiber shrinkage. This was paralleled by a significant increase in hepatic (perivenous) pro-inflammatory CD68+ M1 macrophage and COX2+ cell density (both further enhanced with manifest PDAC), whereby, in fact, significant positive correlations existed between muscular and hepatic proinflammatory CD68+ M1-macrophage or COX2+ cell densities (see Figure 6c,f). Given that decreases in muscle fiber protein content may be detected under conditions of maintained muscle fiber size in early clinical cachexia stages [10], and given that, for example, STAT3-dependent SOCS3 upregulation may lead to protein breakdown [27,50], our present finding of a significant increase in the intramyocellular pool of free proteinogenic amino acids with PanIN, but not PDAC, deserves special consideration. An overall increase in the available pool of proteinogenic amino acids may generally indicate a state of ongoing net-proteolysis, which is also believed to be associated with similarly uniform amino acid efflux from the skeletal muscle. Net-efflux of amino acid from the skeletal muscle, especially of the two marker amino acids phenylalanine or tyrosine, is considered to accurately reflect the net-proteolysis shown by clinical measurements of exchange rates across the leg with catabolic imbalance between measured protein breakdown and synthesis [52,53,54,55]. In support of an assumed proteolytic expansion of the amino acids pool, we found significant positive correlations between Atrogin-1 expression and proteinogenic amino acids (e.g., asparagine: r = 0.603 *p* < 0.001; isoleucine: r = 0.587 *p* < 0.001; histidine: r = 0.581 *p* < 0.001; phenylalanine: r = 0.558 *p* < 0.001 and others) (Figure 8a). Thereby, phenylalanine represented the proteinogenic amino acid pool best (r = 0.812 *p* < 0.001).

Importantly, such a putative net-proteolytic state, present especially with PanIN 2-3 (i.e., a still largely preserved fiber CSA of the gastrocnemius muscle), was associated with an increase in total intramyocellular GSH and a, by trend, enlarged oxidized fraction (GSSG), which indicates an oxidative shift. A proteolytic increase in GSH and its precursor amino acids within the pool of proteinogenic amino acids can be considered as an effective and well-conserved mechanism across species to enhance antioxidant GSH synthesis, in order to rapidly meet oxidative stress within a bidirectional interdependency [56,57]. Here, proteasomal proteolysis is activated even with mild oxidative stress [58]. In fact, we found a significant inverse correlation between Atrogin-1 expression and the total GSH (r = −0.306 *p* < 0.05) and reduced GSH (r = −0.301 *p* < 0.05) in the gastrocnemius muscle. Moreover, significant positive correlations (r > 0.4) existed between the total GSH and pooled proteinogenic (r = 0.447, *p* < 0.01) and its precursor amino acid glycine (r = 0.473, *p* < 0.01) in the gastrocnemius muscle. Thereby, an increased GSH synthesis might have been responsible for the complete intramyocellular consumption of cysteine as a limiting precursor amino acid (generated from methionine in the liver only), because cystine, its disulfide as part of a thiol redox couple, was not detectable in the intramyocellular compartment despite an enlarged amino acid pool.

This metabolic response may tentatively be interpreted as an early compensatory anti-oxidative or -inflammatory reaction following inflammatory cell infiltration and cytokine expression in the skeletal muscles of PanIN mice. In regulatory terms, increased GSH synthesis levels may require induction of the rate of limiting glutamate cysteine ligase (GCL) as part of a general cellular antioxidative response to the observed local inflammatory milieu with PanIN. Among the factors promoting broad antioxidative defense, we presently observed an upregulation of SOCS3 expression, which acts, for example, via the transcription factor nuclear factor erythroid 2-related factor 2 (NRF2) [59]. By binding to the antioxidant response element (ARE), NRF2 responds to mild oxidative stress by inducing the transcription and expression of antioxidant proteins, including the rate limiting enzyme of GSH synthesis and the membrane transporters of GSH precursors, as reported for conditions of increased E3-ligase expression upon fasting [60]. Moreover, NRF2 appears to have an important (age-dependent) role beyond the regulation of antioxidant genes, i.e., in maintaining muscle function contractility, strength, endurance and mass (note that fiber CSA was presently largely unchanged in gastrocnemius muscle of mice with PanIN), as demonstrated in NRF2-deficient mice [61]. Notably, the anti-proteolytic effect of roflumilast, a phosphodiesterase-4 inhibitor, in cachectic COPD patients was associated with upregulation of NRF2 [62]. Thus, the anti-oxidative and -proteolytic role of NRF2 and its upstream signals such as SOCS3 may warrant detailed studies in cancer cachexia, especially its early (precancerous) inflammatory stages, as presently detected.

The improved availability of free amino acids through controlled proteasomal proteolysis via, for example, SOCS3-related E3-ligase expression [50], also includes certain amino acids that reportedly have protein-anabolic signaling potential to the mammalian target of rapamycin (mTOR) and to regulator proteins of translation, such as branched chain amino acids (especially leucine), as well as arginine and glutamine [63,64] or the GSH precursor glycine itself [65]. Significant increases were observed for leucine, valine, proline, serine and alanine in the gastrocnemius muscle in mice with PanIN 1-3 (but only in part with PDAC). It remains unknown whether such increased anabolic amino acid signals may be considered as compensatory (in view of maintained fiber size in PanIN stages) or whether they may lead to an anabolic re-utilization of free amino acids for an altered synthesis program for proteins, such as for anti-oxidative NRF2-induced enzymes or (co)transporters, APR-proteins [27] or others. Notably, the anabolic response, e.g., to leucine as the predominant anabolic stimulus, appears to be unaffected by alterations in SOCS3 expression, as shown at least by SOCS3 deficiency [66].

Thus, one may speculate that with precancerous PanIN conditions, lower-grade local inflammation and oxidative stress may partly be compensated by an increased synthesis of GSH (oxidized to a greater fraction) and by enhanced anabolic amino acid signals to mTOR and other regulators of protein synthesis, resulting in well-maintained fiber CSA. However, as GSH and the (anabolic) amino acid pool decline in the presence of manifest PDAC, muscle fiber size may not be further preserved, as observed in the present study. With PDAC-related aggravated local inflammation, available free amino acids may be increasingly consumed by local APR protein synthesis in the skeletal muscle [27], as well as accelerated hepatic APR, urea cycle and gluconeogenesis, all of which may have contributed to a decrease in intramyocellular amino acid and GSH towards seemingly ‘normal’ values, with a transition from PanIN stages to PDAC. Thereby, decreasing the intracellular availability and oxidative shift in GSH status may also disinhibit pro-apoptotic signals [67], contributing to loss of muscle mass with PDAC, though the data on Casp3, Bax and BCL2 expression with PDAC do not support this notion, at least within the presently studied stage of cachexia.

We also observed a PanIN- and PDAC-related increase in carnosine, a dipeptide composed from histidine and β-alanine with pH-buffering and anti-oxidative capacities. Carnosine is highly concentrated in the skeletal muscle, especially in type 2 fibers, but it is deficient in clinical conditions involving muscle wasting, such as chronic obstructive pulmonary disease (COPD) [68]. Furthermore, carnosine synthesis is limited by β-alanine originating from pyrimidine catabolism and was found to be unrelated to increased levels of histidine.

As a note of caution, the intracellular levels of these free amino acids, carnosine, GSH and GSSG were determined with normalization to total intracellular protein content as commonly done. Changes in protein content may not parallel those in intracellular fluid volume, e.g., under conditions of protein losses, which may theoretically lead to an (uniform) overestimation of amino acid or GSH levels. However, the presently observed changes in amino acids levels were not uniform overall.

In the context of an inflammatory milieu with cachexia, the understudied role of muscular capillarization is of particular interest, as the combination of local muscular inflammation and fiber atrophy has also been reported for chronic hypoxic exposure [69,70]. Increases in capillary density with chronic hypoxia are mainly attributable to fiber atrophy, however, they obviously require increased local VEGF expression which contributes to unchanged capillary-to-fiber-ratio. In contrast, a VEGF upregulation was absent in this study, at least in the soleus muscle, and capillary density was unaltered not only in mice with PanIN, but also in mice with PDAC, despite significant fiber atrophy (i.e., increased number of fibers per area) in line with previous clinical findings of our group [71]. As a measure more relevant to fiber O_2_ supply, we also calculated the number of capillary contacts per unit fiber CSA, which, despite significant local inflammation, was unaltered with PanIN or PDAC. Obviously, the connection between fiber size and histological fiber-adjacent capillary supply may be maintained with cachexia, at least in the current stage of study.

Though limited to subgroups and to the fast-twitch quadriceps muscle resembling the gastrocnemius muscle, our data provide preliminary evidence that the development of PanIN and, more importantly, of PDAC as a cause of wasting largely occurs without alterations of NMJ density or integrity, as determined by absolute areas of presynapsis and postsynapsis, or by their absolute or percentage overlap. However, as a note of caution, our data reveal a considerable trend towards increased percentage pre-/postsynaptic overlap (Table 6) which warrants larger studies, as denervation or NMJ degeneration may not generally be associated with decreased postsynaptic NMJ volume or area, but in some cases with enlarged NMJ area [72,73]. Evidence against the role of denervation in cachexia-related muscle wasting arises also from a recent clinical study showing that NMJ remained structurally intact in the non-locomotor rectus abdominis muscle of gastric cancer patients with fiber atrophy and weight loss, compared to those without [6,39]. This may contrast with some indirect evidence from C26-mice for cachexia-related denervation arising from increased nCAM expression and centronucleation [38], presently not observed with PDAC. The role of denervation as an important cause of muscle atrophy thus warrants further dedicated studies, especially, since muscle wasting with COPD reveals multiple similarities with cancer cachexia and is now considered to involve deterioration of NMJ [74].

Moreover, our preliminary data from subgroups also speak against ultrastructural mitochondrial damage with PDAC, though lower mitochondrial content and respiratory rates have been described to occur with cancer [6,40,75]. PGC1α, a relevant signal for mitochondriogenesis and fiber differentiation, was found to be unaltered in the soleus muscle in the present study, as was p62 as an index of autophagic activity required for mitochondrial turnover/renewal. However, muscle specific SOCS3 (over)expression may lead to the inhibition of mitochondrial gene expression, as well as structural alterations [28]. Thus, any manifest ultrastructural damage may be preceded by mitochondrial (membrane) dysfunction, which may facilitate pro-apoptotic signals.

Limitations: The present study, using a generally advantageous cachexia model of PDAC, did not assess relevant factors of muscle mass and function such as tumor mass [76] and staging including ascites, metastasis, edema, calory intake or physical activity. Though pancreas histopathology was assessed by two independent, experienced investigators, ultimately, it cannot be excluded that the 3x-transgenic mice assigned to the group of tumor-free PanIN 2-3 stages may have had early PDAC in areas not examined by serial cross-sections. However, it is unlikely that the 2x-transgenic mice, especially those with PanIN 1A-B, may have had invasive PDAC with a delayed and low incidence of 6%. As another limitation, muscle (wet) weight, i.e., losses in muscle mass in total could not be quantified, as immediate shock freezing was required for reliable metabolic measurements. It therefore remains unknown, whether, alongside significant PDAC-related fiber atrophy, a reduction in fiber count through apoptosis may also have contributed to muscle wasting. Notably, soleus muscle pax7 and myogenin expressions were unaltered with PDAC, indicating unaffected myogenesis, despite the increased endomysial inflammatory macrophage infiltration known to be involved in satellite cell activation for muscle fiber regeneration.

Obviously, our data pointed towards a considerable impact of gender beside PDAC on fiber size, expressions of Atrogin-1, Casp3, Bax, CD68, TNFα or MMP9 in the gastrocnemius muscle and of p61, CD68 or myogenin in the soleus muscle, as well as on the majority of intracellular proteinogenic amino acids or the dipeptide carnosine and its precursor histidine. The small male or female sample size qualified our explorative analyses as hypothesis-generating, warranting a larger investigation on the impact of gender, beside age, on cachexia. Notably, the present comparison of PDAC or PanIN stages to WT mice does not easily translate into a clinical comparison between cachectic vs. non- or pre-cachectic cancer patients, which may represent different time windows of cachexia development through PDAC. As a more general remark, our study was unable to represent the complex clinical conditions of progressive cancer cachexia when dealing with a higher age group, comorbidities, physical inactivity, pain or nutritional and psychosocial problems as imposed with various tumor entities, stages, treatments and complications.

## 5. Conclusions

The present study on KPC mice demonstrated that the onset of morphological muscle wasting in ‘white’ and ‘red’ skeletal muscles clearly coincides with the manifestation of PDAC, but not mutant-based precancerous PanIN stages. It was also associated with the partial upregulation of E3 ligases and local inflammation in terms of the endomysial infiltration of CD68+ and COX2+ cells (macrophages), as well as increased myocellular expression of SOCS3 and cytokines. The data suggest the impact of gender and muscle type, thus warranting further analysis. However, they appear to exclude alterations in the mitochondria, capillarization or NMJ morphology as initial triggers of muscle wasting. Surprisingly, significant macrophage infiltration and massive SOCS3 upregulation were already detectable with PanIN 1A-B, together with evidence for substantial, likely proteolytic, increases in the intramyocellular free amino pool and an increase in the total GSH. This may tentatively be interpreted as an initial defense response to local inflammatory/oxidative stress that may be overridden by further (hepatic or other) amino acids consumption with PDAC, resulting in fiber atrophy. Our data warrant further studies on ‘pre-cachectic’ states that might affect muscle (protein) metabolism and/or function before the manifestation of massive fiber atrophy as indicated by a decreased myocellular protein content or function at unaltered fiber size [10,77]. Overall, early inflammatory and related metabolic changes in the liver [4,12,78,79] occurring downstream of pancreatic inflammation [13,32,33,80] should be monitored in parallel to those in the skeletal muscles to delineate the initial steps of (pre)cachexia, especially in PDAC and other gastrointestinal malignancies with portal venous drainage and high incidences and degrees of cachexia. Notably, liver macrophages have been implicated in immune surveillance and the amplification of the immune response, e.g., in COPD associated with muscle wasting [81].

## Figures and Tables

**Figure 1 cells-11-01607-f001:**
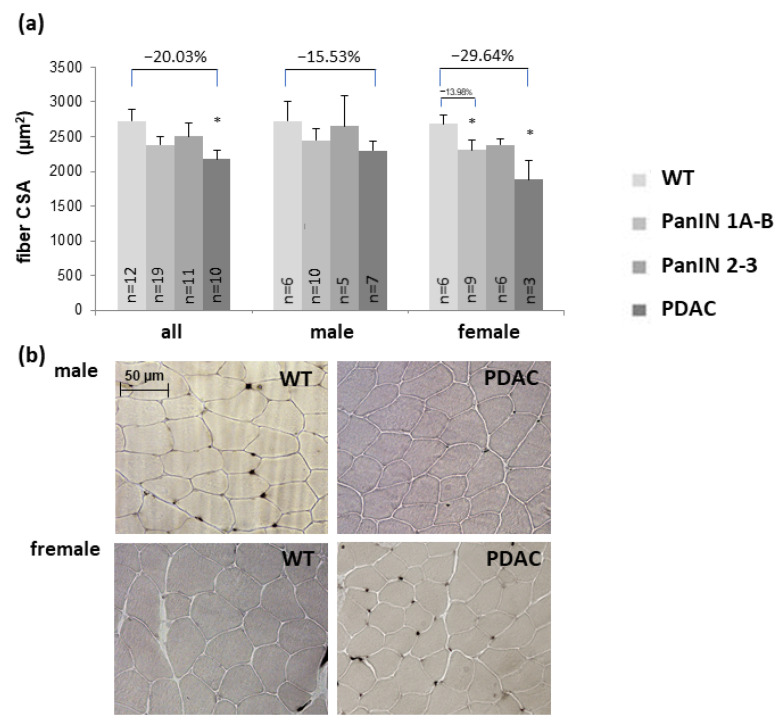
(**a**) Cross-sectional area (CSA) of gastrocnemius muscle type 2x fibers in the four groups WT, PanIN 1A-B, PanIN 2-3, and PDAC within the total study population of males, or females as indicated. Percentage differences exceeding 10% are indicated at the top. Mean ± SEM, * for *p* < 0.05 compared to WT by Student’s *t*-test for unpaired observation. For two-factorial ANOVA controlling for factors PDAC, sex and age see text in Results section. (**b**) Representative male and female examples of ATPase staining for the WT and PDAC.

**Figure 2 cells-11-01607-f002:**
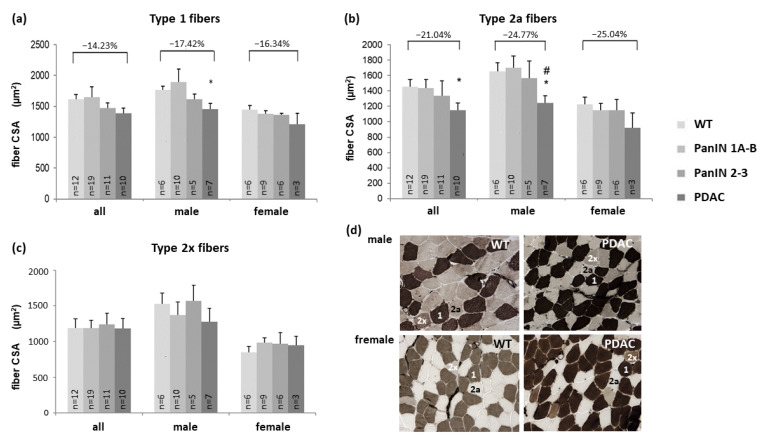
Cross-sectional area (CSA) of soleus muscle (**a**) type 1, (**b**) type 2a, (**c**) type 2x fibers in the four groups WT, PanIN 1A-B, PanIN 2-3, and PDAC. Each panel (**a**–**c**) presents data for the total study population, males, or females as indicated. Percentage differences exceeding 10% are indicated on top of each graph. Mean± SEM, * for *p* < 0.05 compared to WT, # for *p* < 0.05 compared to PanIN 2-3 by Student’s *t*-test for unpaired observation. For two-factorial ANOVA controlling for factors PDAC, sex and age see text in Results section; this approach also detected a significant CSA-decreasing effect of PDAC on type 2x fibers. (**d**) Representative male and female examples of ATPase staining for the WT and PDAC group with identification of fiber types 1, 2a, and 2x. *p* = 0.010, age: *p* = 0.565) in soleus muscle.

**Figure 3 cells-11-01607-f003:**
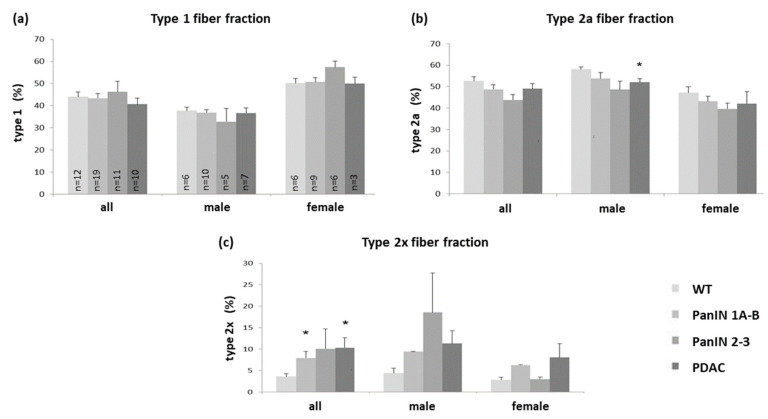
Fiber distribution (% fraction) of (**a**) type 1 (**b**) type 2a and (**c**) type 2x fibers in soleus muscle in the four groups WT, PanIN 1A-B, PanIN 2-3, and PDAC, with each panel (**a**–**c**) presenting data for the total study population, males or females as indicated. Mean ± SEM, * for *p* < 0.05 compared to WT.

**Figure 4 cells-11-01607-f004:**
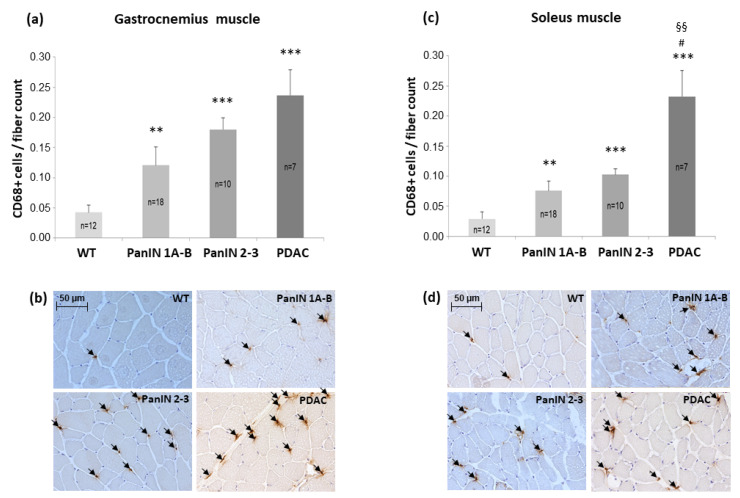
Mean density of CD68+ macrophages (count/muscle fiber count) in (**a**) gastrocnemius muscle with (**b**) representative immunolocalizations of CD68+ macrophages (see arrows) and in (**c**) soleus muscle with (**d**) representative immunolocalizations of CD68+ macrophages (see arrows) within the four groups WT, PanIN 1A-B, PanIN 2-3, and PDAC as indicated. Mean ± SEM, ** for *p* > 0.01 and *** for *p* > 0.001 vs. WT, # for *p* < 0.05 vs. PanIN 2-3, and §§ for *p* < 0.01 vs. PanIN 1A-B by Student’s *t*-test or Mann–Whitney U test (posthoc) for unpaired observation.

**Figure 5 cells-11-01607-f005:**
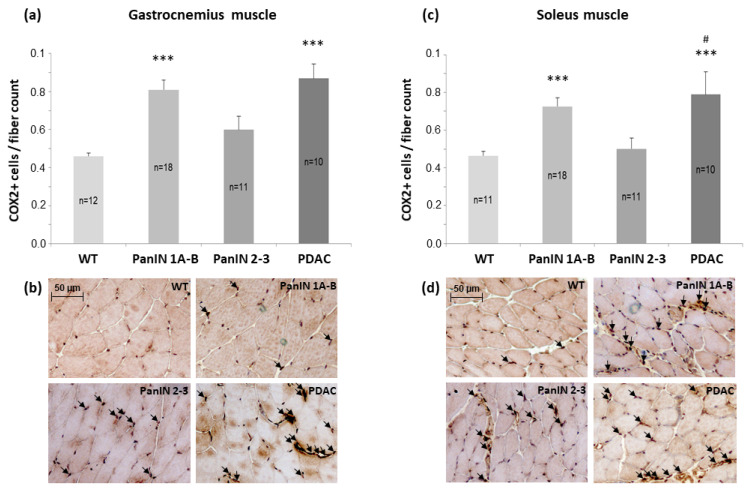
Mean density of COX2+ cells (count/muscle fiber count) in (**a**) gastrocnemius muscle with (**b**) representative immunolocalizations of COX2+ cells (see arrows) and in (**c**) soleus muscle with (**d**) representative immunolocalizations of COX2+ cells (see arrows) within the four groups WT, PanIN 1A-B, PanIN 2-3, and PDAC as indicated. Mean ± SEM, *** for *p* > 0.001 vs. WT, # for *p* < 0.05 vs. PanIN 2-3 by Student’s *t*-test or Mann–Whitney U test (posthoc) for unpaired observation.

**Figure 6 cells-11-01607-f006:**
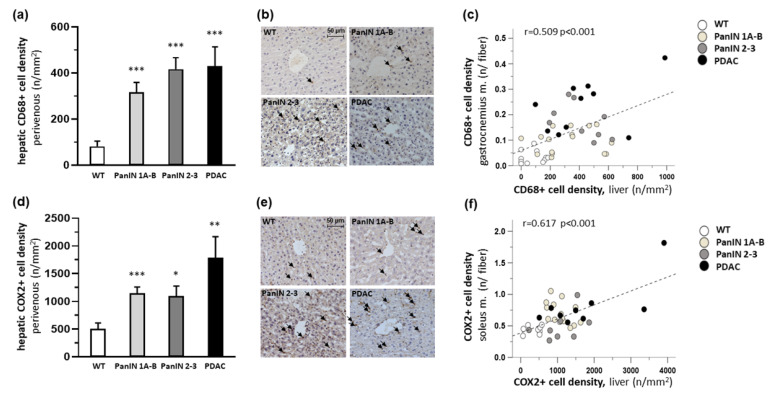
Mean density of inflammatory cells in the liver within the four groups WT, PanIN 1A-B, PanIN 2-3, and PDAC: (**a**) perivenous CD68+ macrophage density (count/area); (**b**) representative immunolocalizations of CD68+ macrophages (see arrows); (**c**) significant positive correlation between CD68+ macrophage density in gastrocnemius muscle and liver; (**d**) perivenous COX2+ cell density (count/area); (**e**) representative immunolocalizations of COX2+ cells (see arrows); (**f**) significant positive correlation between COX2+ cell density in soleus muscle and liver. (**a**,**d**) show mean ± SEM, * for *p* < 0.05, ** for *p* > 0.01 and *** for *p* > 0.001 vs. WT.

**Figure 7 cells-11-01607-f007:**
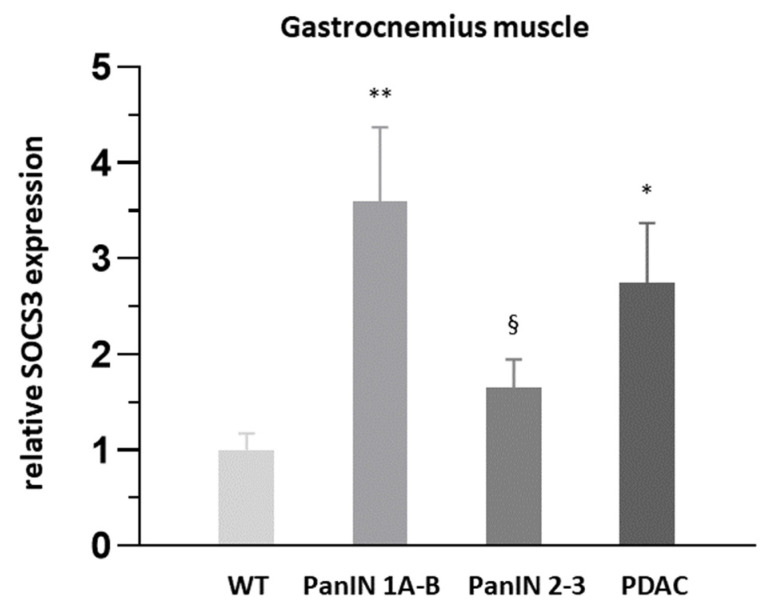
Relative x-fold SOCS3 expression in gastrocnemius muscle with PanIN 1A-B, PanIN 2-3 and PDAC compared to WT. Mean ± SEM, * for *p* < 0.05, ** for *p* > 0.01 vs. WT, § for *p* < 0.01 vs. PanIN 1A-B by Student’s *t*-test or Mann Whitney U test (posthoc) for unpaired observation.

**Figure 8 cells-11-01607-f008:**
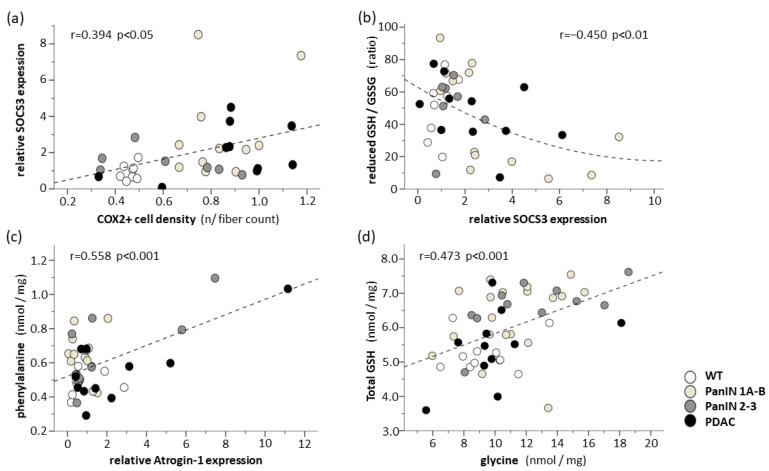
Correlation between (**a**) relative SOCS3 expression and COX2+ cell density, (**b**) reduced GSH to GSSG ratio and relative SOCS3 expression, (**c**) intramyocellular phenylalanine level and relative atrogin-1 expression, as well as (**d**) total intramyocellular GSH and glycine levels. Scatterplots with indication of groups WT, PanIN 1A-B, PanIN 2-3 and PDAC, Pearson’s correlation coefficient, regression line (non-linear in case of (**b**)) and *p* value.

**Table 1 cells-11-01607-t001:** Pancreatic phenotype, genotype, age and body weight of the total, male and female study population assigned to the WT, PanIN 1A-B, PanIN 2-3 or PDAC mice.

Pancreatic Phenotype		WT (Normal)	PanIN 1A-B	PanIN 2-3	PDAC
*n*	all	13	22	11	11
	male	7	12	5	7
	female	6	10	6	4
Genotype	all	13 WT	16 2x-, 6 3x-transgenic	6 2x-, 5 3x transgenic	11 3x-transgenic
	male	7 WT	10 2x-, 2 3x-transgenic	3 2x-, 2 3x-transgenic	7 3x-trasngenic
	female	6 WT	6 2x-, 4 3x-transgenic	3 2x-, 3 3x-transgenic	4 3x-transgenic
Age (weeks)	all	17.0 ± 0.6	21.7 ± 1.8	20.4 ± 1.2 **	23.6 ± 1.0 ***#
	male	17.1 ± 1.0	22.4 ± 2.6	20.2 ± 2.2	24.3 ± 1.3 **
	female	16.8 ± 0.5	20.9 ± 2.4	20.5 ± 1.3 *	22.5 ± 1.6 *
Body weight (g)	all	29.0 ± 1.7	27.5 ± 1.2	27.4 ± 1.7	27.7 ± 1.0
	male	34.0 ± 1.5	31.2 ± 1.3	31.8 ± 2.7	29.8 ± 0.4 *
	female	23.2 ± 0.6 $$$	23.2 ± 0.5 $$$	23.8 ± 0.6 $	24.1 ± 1.5 $$

Mean ± SEM. Two-factorial ANOVA detected significant impact of sex (*p* < 0.001) but not of PDAC (*p* = 0.194) or of PanIN (*p* = 0.374). However, PDAC (*p* = 0.002) beside and sex (*p* < 0.001) significantly impacted body weight when controlled for age (*p* = 0.146). Further, the combined PanIN stages (*p* = 0.002) beside sex (*p* < 0.001) significantly affected body weight with control age (*p* < 0.001). * *p* < 0.05, ** *p* < 0.01, and *** *p* < 0.001 Student’s *t*-test or Mann–Whitney U test compared to WT (normal pancreatic phenotype) mice; # *p* < 0.05 compared to PanIN 2-3. $ *p* < 0.05, $$ *p* < 0.01, and $$$ *p* < 0.001 comparing female vs. male body weight. The 2x-trangenic and the 3xtransgenic genotypes refer to the Kras-PDX mutant and the Kras-p53-PDX-mutant, respectively.

**Table 2 cells-11-01607-t002:** Histomorphometry of fiber type-specific capillarization of gastrocnemius and soleus muscle in WT, PanIN 1A-B, PanIN 2-3, or PDAC mice.

KERRYPNX		WT	PanIN 1AB	PanIN 2-3	PDAC	*p* Value ANOVA		*p* Value ANOVA	
						PDAC	Sex	PanIN	Sex
*n* (m/f)		12–13 (6–7/6)	16–18 (8–10/8)	9–11 (4–5/5–6)	10 (7/3)	22–23 (13–14/9)		37–42 (18–22/19–20)	
Gastrocnemius muscle (type 2x fiber)									
Capillary density	mm^−2^	70.4 ± 3.4	76.9 ± 6.2	70.4 ± 6.5	70.3 ± 4.2	0.846	0.542	0.599	0.794
Capillary/fiber	ratio	1.29 ± 0.11	1.29 ± 0.06	1.15 ± 0.09	1.15 ± 0.05	0.210	0.342	0.655	0.463
Capillary contacts	*n*	2.94 ± 0.20	2.80 ± 0.16	2.82 ± 0.13	2.56 ± 0.18	0.210	0.415	0.730	0.226
Capillaries/CSA	*n*/10^3^ µm^2^	1.35 ± 0.20	1.22 ± 0.13	1.22 ± 0.11	1.06 ± 0.07	0.203	0.255	0.459	0.600
Soleus muscle (type 1, 2a and 2x fibers)									
Capillary density	mm^−2^	194 ± 12	194 ± 15	184 ± 13	196 ± 12	0.864	0.487	0.916	0.549
Capillary/fiber	ratio	2.09 ± 0.20	2.24 ± 0.14	1.98 ± 0.24	1.75 ± 0.14	0.119	0.078	0.827	0.121
Capill. contacts									
fiber type 1	*n*	4.77 ± 0.33	4.71 ± 0.25	4.78 ± 0.26	4.37 ± 0.19	0.534	0.285	0.818	0.036
fiber type 2a	*n*	4.40 ± 0.26	4.39 ± 0.26	4.50 ± 0.25	4.21 ± 0.21	0.864	0.295	0.693	0.009
fiber type 2x	*n*	3.89 ± 0.50	4.39 ± 0.34	3.80 ± 0.43	4.00 ± 0.26	0.657	0.142	0.334	0.225
Capillaries/CSA									
fiber type 1	*n*/10^3^ µm^2^	4.06 ± 0.91	3.42 ± 0.25	3.68 ± 0.30	3.28 ± 0.23	0.394	0.548	0.403	0.094
fiber type 2a	*n*/10^3^ µm^2^	4.20 ± 0.83	3.41 ± 0.34	3.92 ± 0.39	3.73 ± 0.35	0.751	0.121	0.366	0.063
fiber type 2x	*n*/10^3^ µm^2^	3.54 ± 0.37	3.71 ± 0.38	3.86 ± 0.34	3.65 ± 0.39	0.706	0.541	0.668	0.230

Mean ± SEM. For *p* values by univariate ANOVA considering factors PDAC and sex, or PanIN and sex (see right side). A significant impact of sex was detected with capillary contacts of soleus type 1 and 2a fibers of PanIN mice. No significant interaction of PDAC/PanIN with sex and no significant impact of age was detected (see Section 2.10). No significant differences were detected by Student’s *t*-test (posthoc).

**Table 3 cells-11-01607-t003:** (**a**) Relative gene expression in gastrocnemius muscle of WT, PanIN 1A-B, PanIN 2-3, or PDAC mice. (**b**) Relative gene expression in soleus muscle with WT and PDAC.

**(a) gastrocenmius muscle**								
	**WT**	**PanIN 1A-B**	**PanIN 2-3**	**PDAC**	***p* Value ANOVA**		***p* Value ANOVA**	
					**PDAC**	**Sex**	**PanIN**	**Sex**
*n* (m/f)	9–11 (5–6/4–5)	17 (10/7)	10 (4/6)	11 (7/4)	20–22 (12–13/8–9)		36–38 (19–20/17–18)	
Atrophy and apoptosis signals								
Atrogin-1	1 ± 0.35	0.83 ± 0.14	2.12 ± 0.90	2.80 ± 1.08	0.065	0.103	0.664	0.903
MuRF1	1 ± 2.45			0.14 ± 0.16	0.188	0.388		
Casp3	1 ± 0.26	1.44 ± 0.26	1.14 ± 0.23	1.29 ± 0.23	0.127	0.001	0.287	0.043
Bax	1 ± 0.41	0.70 ± 0.11	1.75 ± 0.68	1.65 ± 0.58	0.226	0.030	0.935	0.142
BCL2	1 ± 0.18	0.91 ± 0.14	1.64 ± 0.53	1.44 ± 0.26	0.279	0.743	0.625	0.429
Pro-/anti-inflammatory signals and oxidative stress								
CD68	1 ± 0.39	1.95 ± 0.85	0.82 ± 0.24	2.56 ± 0.82	0.046	0.045	0.554	0.191
COX2	1 ± 0.33	2.58 ± 0.55 *	2.00 ± 0.54	3.30 ± 1.44	0.264	0.984	0.122	0.632
IL1β	1 ± 0.63	0.99 ± 0.18	2.07 ± 1.07	3.52 ± 1.36 *	0.089	0.078	0.797	0.541
IL6	1 ± 0.52	0.68 ± 0.16	0.97 ± 0.46	1.65 ± 0.47	0.192	0.263	0.652	0.957
TNFα	1 ± 0.26	2.34 ± 0.59 *	1.24 ± 0.47	1.84 ± 0.63	0.040	0.027	0.221	0.456
SOCS3	1 ± 0.17	3.60 ± 0.77 **	1.65 ± 0.29	2.74 ± 0.63 *	0.050	0.886	0.038	0.690
MAO-A	1 ± 0.36	0.39 ± 0.06	0.98 ± 0.45	0.99 ± 0.29	0.877	0.152	0.210	0.384
MAO-B	1 ± 0.42	0.39 ± 0.06	1.46 ± 0.61	0.93 ± 0.37	0.948	0.164	0.691	0.336
MMP9	1 ± 0.35	0.67 ± 0.13	1.71 ± 0.94	1.72 ± 0.52	0.126	0.026	0.980	0.605
PGC1α	1 ± 0.51	0.36 ± 0.06	0.65 ± 0.31	0.82 ± 0.16	0.750	0.749	0.154	0.571
**(b) soleus muscle**								
	**WT**		**PDAC**		***p* Value ANOVA**			
					**PDAC**		**Sex**	
*n* (m/f)	11 (6/5)		7 (5/2)		18 (11/7)			
Atrophy and apoptosis signals								
Atrogin-1	1 ± 0.36		1.25 ± 0.73		0.420		0.796	
MuRF1	1 ± 0.71		1.70 ± 0.91		0.097		0.743 !	
Casp3	1 ± 0.47		0.92 ± 0.37		0.508		0.078	
Bax	1 ± 0.22		0.90 ± 0.07		0.345		0.953	
BCL2	1 ± 0.52		0.94 ± 0.53		0.636		0.145	
p62	1 ± 0.16		1.19 ± 0.34		0.211		0.033	
Pro-/anti-inflammatory signals and oxidative stress								
CD68	1 ± 0.24		1.40 ± 0.49 *		0.033		0.019 !	
COX2	1 ± 1.09		2.55 ± 2.08		0.104		0.682	
IL1β	1 ± 0.65		7.08 ± 5.65 **		0.005		0.282	
IL6	1 ± 0.34		1.18 ± 0.63		0.513		0.780	
TNFα	1 ± 0.81		1.16 ± 1.09		0.893		0.267	
SOCS3	1 ± 1.12		2.32 ± 1.89		0.147		0.093	
MAO-A	1 ± 0.21		0.94 ± 0.25		0.798		0.223	
MAO-B	1 ± 0.23		0.89 ± 0.13		0.345		0.602	
MMP9	1 ± 0.61		0.63 ± 0.36		0.268		0.514	
PGC1α	1 ± 1.02		1.74 ± 1.05		0.272		0.122	
Myogenic signals								
MyoG	1 ± 0.41		0.92 ± 0.22		0.983		0.004	
pax7	1 ± 0.62		1.03 ± 0.59		0.877		0.177	
Angiogenic signal								
VEGFA	1 ± 0.24		11.4 ± 0.21		0.378		0.080	

(a) gastrocnemius muscle: Mean ± SEM. For *p* values by univariate ANOVA considering factors PDAC and sex, or PanIN and sex see right side. * *p* < 0.05, and ** *p* < 0.01 Student’s *t*-test or Mann–Whitney U test (posthoc) compared to WT mice. Gene expressions were normalized for housekeeper b-actin in gastrocnemius muscle and presented relative to WT mice. (b) soleus muscle: Mean ± SEM. For *p* values by univariate ANOVA considering factors PDAC and sex see right side. ! *p* < 0.05 for significant interaction. * *p* < 0.05 and ** *p* < 0.01 by Student’s *t*-test or Mann–Whitney U test (posthoc) compared to WT. Gene expressions were normalized for housekeeper RER1 for soleus muscle and presented relative to WT.

**Table 4 cells-11-01607-t004:** Intramyocellular pool of free proteinogenic amino acids including branched-chain and other amino acids with anabolic signaling potential in WT, PanIN 1A-B, PanIN 2-3 or PDAC mice.

	WT	PanIN 1A-B	PanIN 2-3	PDAC	*p* Value ANOVA		*p* Value ANOVA	
					PDAC	Sex	PanIN	Sex
*n* (m/f)	11 (6/5)	19 (10/9)	11 (5/6)	11 (7/4)	22 (13/9)		41 (21/20)	
Proteinogenic amino acids (nmol/mg)	51.18 ± 1.94	59.13 ± 2.36 *	69.12 ± 4.85 **	55.84 ± 3.51 #	0.075	0.004	0.006	0.022
Leucine (nmol/mg)	0.802 ± 0.045	1.133 ± 0.055 ***	1.338 ± 0.121 **	1.053 ± 0.112	0.006	0.004	0.000	0.112
Isoleucine (nmol/mg)	0.406 ± 0.041	0.485 ± 0.030	0.519 ± 0.069	0.503 ± 0.049	0.055	0.016	0.138	0.567
Valine (nmol/mg)	0.953 ± 0.053	1.252 ± 0.055 **	1.387 ± 0.087 **	1.221 ± 0.080 *	0.010	0.097	0.000	0.248
Arginine (nmol/mg)	1.195 ± 0.175	1.254 ± 0.176	1.804 ± 0.318	1.262 ± 0.244	0.221	0.000	0.376	0.000
Glutamine (nmol/mg)	7.660 ± 0.466	8.383 ± 0.411	10.410 ± 1.051 *	9.079 ± 0.836	0.023	0.002	0.091	0.031
Serine (nmol/mg)	1.486 ± 0.092.	1.671 ± 0.101	2.118 ± 0.179 **	1.451 ± 0.143 ##	0.692	0.082	0.043	0.528
Lysine (nmol/mg)	3.469 ± 0.577	3.557 ± 0.514	5.654 ± 0.926	4.135 ± 1.102	0.182	0.000	0.288	0.000
Proline (nmol/mg)	1.266 ± 0.092	1.623 ± 0.073 **	1.945 ± 0.182 **	1. 537 ± 0.144	0.117	0.959	0.003	0.567
Alanine (nmol/mg)	10.96 ± 0.39	13.94 ± 0.55 **	16.01 ± 0.92 ***	12.65 ± 0.86 #	0.089	0.723	0.000	0.709

Mean ± SEM; for *p* values by univariate ANOVA considering factors PDAC, sex (and age), or PanIN, sex (and age) see right side. No significant impact of the factor age (*p* values not shown) and no significant interactions were detected (see Section 2.10). * *p* < 0.05, ** *p* < 0.01, and *** *p* < 0.001 Student’s *t*-test (posthoc) vs. WT mice # *p* < 0.05, and ## *p* < 0.01 Student’s *t*-test compared to PanIN 2-3. Intramyocellular proteinogenic amino acids do not include cystine and tryptophane because they were not detectable.

**Table 5 cells-11-01607-t005:** Intramyocellular antioxidants glutathione and carnosine as well as their (proteinogenic) precursor amino acid in WT, PanIN 1A-B, PanIN 2-3, or PDAC mice.

	WT	PanIN 1A-B	PanIN 2-3	PDAC	*p* Value ANOVA		*p* Value ANOVA	
					PDAC	Sex	PanIN	Sex
*n* (m/f)	11 (6/5)	21 (12/9)	11 (5/6)	11 (7/4)	22 (13/9)		43 (23/20)	
GSH total (nmol/mg)	5.24 ± 0.16	6.42 ± 0.23 ***	6.62 ± 0.23 ***	5.45 ± 0.32 ##	0.731	0.130	0.000	0.317
GSH reduced (nmol/mg)	5.12 ± 0.16	6.17 ± 0.22 **	6.39 ± 0.25 ***	5.29 ± 0.32 #	0.796	0.183	0.001	0.335
GSSG (nmol/mg)	0.117 ± 0.019	0.247 ± 0.048 *	0.229 ± 0.050	0.161 ± 0.052	0.621	0.287	0.058	0.808
GSH reduced/GSSG	60.1 ± 13.1	42.3 ± 5.6	41.5 ± 6.6	47.7 ± 6.1	0.535	0.737	0.106	0.643
Cystine (nmol/mg)	n.d.	n.d.	n.d.	n.d.				
Glutamate (nmol/mg)	4.03 ± 0.25	3.79 ± 0.28	3.65 ± 0.42	3.42 ± 0.32	0.162	0.006	0.389	0.014
Glycine (nmol/mg)	9.40 ± 0.60	11.03 ± 0.60	12.39 ± 1.06 *	10.08 ± 0.92	0.583	0.995	0.041	0.755
Carnosine (nmol/mg)	12.45 ± 2.00	15.35 ± 1.94	13.30 ± 2.06	16.45 ± 2.18	0.024	0.000	0.049	0.000
Histidine (nmol/mg)	0.510 ± 0.029	0.600 ± 0.026 *	0.751 ± 0.085 *	0.629 ± 0.060	0.007	0.011 !	0.032	0.472

Mean ± SEM, n.d. = not detected; * *p* < 0.05, ** *p* < 0.01, and *** *p* < 0.001 Student’s *t*-test compared to WT mice. # *p* < 0.05, and ## *p* < 0.01 Student’s *t*-test compared to PanIN 2-3. ! *p* < 0.05, for interaction. Note that β-alanine, the second amino acid forming the dipeptide carnosine beside histidine, was not measured.

**Table 6 cells-11-01607-t006:** Morphology of NMJ pre- and post-synapsis in quadriceps muscle in WT, PanIN 1A-B, PanIN 2-3 or PDAC mice.

	WT	PanIN 1A-B	PanIN 2-3	PDAC
*n* (m/f)	5 (3/2)	5 (2/3)	5 (2/3)	4 (4/0)
Btx+ NMJ density [mm^2^]	29.1 ± 7.5	32.8 ± 5.7	44.1 ± 13.2	24.9 ± 4.9
Btx+ area [µm^2^]	2.85 ± 0.18	2.88 ± 0.22	2.76 ± 0.11	3.06 ± 0.40
vAChT+ area [µm^2^]	3.12 ± 0.28	3.03 ± 0.36	3.41 ± 0.36	4.31 ± 0.58
vAChT—Btx Overlap [µm^2^]	0.53 ± 0.03	0.56 ± 0.10	0.61 ± 0.12	0.98 ± 0.25
vAChT—Btx Overlap [%]	18.61 ± 0.57	18.67 ± 2.85	21.96 ± 2.34	30.54 ± 4.20

Mean ± SEM; No significant differences were detected between PDAC or PanIN stages and WT for any of the parameter. Due to small group size, the impact of sex was not controlled for by ANOVA.

## Data Availability

The data sets used and/or analyzed during the current study are available from the corresponding author on reasonable request.

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
