# Peer review of "Inflammation and Wasting of Skeletal Muscles in Kras-p53-Mutant Mice with Intraepithelial Neoplasia and Pancreatic Cancer—When Does Cachexia Start?"

_cells, 2022, doi:10.3390/cells11101607_

Round 1

Reviewer 1 Report

Comments

  1. Spelling error in title, “intraepithelial”? The title needs to be modified, it may be “…muscles in Kras-p53-mutant mice with intraepithelial neoplasia and pancreatic cancer…..”.
  2. The grammar in this article needs to be edited. It is very hard to read.
  3. The organization of table and figure need to follow the description in Results.

line 22-37 That is not easy to understand what the authors want to describe. 

Author Response

Manuscript ID  1653643    Hildebrandt, Keck et al.  ’’Inflammation and wasting of skeletal muscles … ’’

Point-to-Point Response to the Reviewers’ comments

Dear Editor, 

We would like to thank the Reviewers for their time and valuable comments which were carefully considered for improving this manuscript. Please find below our Point-to-Point response to each of the Reviewers’ comments.

All changes are marked up by yellow highlighting within the text body and legends.

Please note, that the edited manuscript version (returned for revision by CELLS) had line numbers only in the Discussion (!). We were not quite sure if this was intended (or just an error), however, we decided to identify our major changes by page/paragraph/line informations in addition to yellow highlighting.

As pointed out Reviewer 1 and Reviewer 3, the order of Tables and Figures had to be rearranged, as it did not follow the results in this returned MDPI-CELLS edited version. For example, Table 1 now precedes Figure 1 and 2 (again) in line with presentation order or results.

We would like to point out, that the old (submitted) Figure 1 is now divided into two separate Figures, i.e. a new Figure 1 (gastrocnemius muscle) and a new Figure 2 (soleus muscle) to enable the inclusion of representative examples (for male and female mice) as requested by Reviewer 2.

All other Figures were renumbered and the Legends of Figure 1 and 2 rewritten accordingly.

In Figure 4, we now chose better examples of gastrocnemius CD68+ immunolocalization to improve discrimination between CD68+ macrophages and fiber nuclei (hematoxylin).

We now have also included a missing Reference (new Ref. Nr. 41) regarding pancreatic histopathology criteria and changed the consecutive Reference numbers accordingly.

Regarding the layout of Tables 1-6 in the present MDPI-CELLS layout, obvious problems exist with the alignments of parameters, dimensions and of mean values ±SEM as well as the line spacing appear (partly disrupted). We trust, that this will be solved /optimized in the editing process according to the journal style of CELLS.

We hope that our revised manuscript is now adequate for publication in CELLS.

With kind regards,

Sincerely

Wulf Hildebrandt (corresponding author)

  • Reviewer

Open Review

English language and style

(x) Extensive editing of English language and style required
( ) Moderate English changes required
( ) English language and style are fine/minor spell check required
( ) I don't feel qualified to judge about the English language and style

Yes

Can be improved

Must be improved

Not applicable

Does the introduction provide sufficient background and include all relevant references?

( )

( )

(x)

( )

Is the research design appropriate?

( )

( )

( )

(x)

Are the methods adequately described?

( )

( )

(x)

( )

Are the results clearly presented?

( )

( )

(x)

( )

Are the conclusions supported by the results?

( )

( )

(x)

( )

Comments and Suggestions for Authors

Comments

  1. Spelling error in title, “intraepithelial”? The title needs to be modified, it may be “…muscles in Kras-p53-mutant mice with intraepithelial neoplasia and pancreatic cancer…..”.
  2. The grammar in this article needs to be edited. It is very hard to read.
  3. The organization of table and figure need to follow the description in Results.
  1. line 22-37 That is not easy to understand what the authors want to describe. 

Response:

Ad 1. Changed as suggested, i.e. corrected to ‚intraepithelial‘ and the word order changed to  “Inflammation and wasting of skeletal muscles in Kras-p53-mutant mice with intraepithelial neoplasia and pancreatic cancer – when does cachexia start ?”.

Ad 2. As a native English speaker, the co-author E. P. Slater has once again checked the language/style, however, we certainly agree to language/style editing upon the editor’s request.

Ad 3. We totally agree and, thus, have rearranged the order of tables and figures such that they now clearly follow the descriptions of the Results section. Please note that we have now divided the former Figure 1 into two separate new Figures, i.e. new Figure 1 (gastrocnemius muscle) and new Figure 2 (soleus muscle), which both now include representative ATPase staining examples for a male and a female mouse of the WT and PDAC group. Please see new Figure 1 (page 10) and new Figure 2 (page 11)

Ad 4. Unfortunately, we cannot identify line 22-37 in this returned version, as line numbers exists in Discussion only (likely due to some technical error). Could you please specify?

  • Reviewer

Open Review

English language and style

( ) Extensive editing of English language and style required
( ) Moderate English changes required
(x) English language and style are fine/minor spell check required
( ) I don't feel qualified to judge about the English language and style

Yes

Can be improved

Must be improved

Not applicable

Does the introduction provide sufficient background and include all relevant references?

(x)

( )

( )

( )

Is the research design appropriate?

(x)

( )

( )

( )

Are the methods adequately described?

(x)

( )

( )

( )

Are the results clearly presented?

(x)

( )

( )

( )

Are the conclusions supported by the results?

(x)

( )

( )

( )

Comments and Suggestions for Authors

Dear Authors,

Cancer Cachexia is a complex metabolic syndrome commonly associated with presence of systemic inflammation. The presented article is addressing an important question whether muscular inflammation plays a role in muscle wasting. The subject is extremely important for early understanding of cachexia mechanisms. However, minor comments/revisions were made bellow to improve the quality of the present publication.

  1. In the figure 1 you provided information about reduction in skeletal muscle fiber CSA in PDAC mice compared with WT. Could you please provide morphological analysis showing representative images to support these findings?
  2. CD68 and COX2 results in muscle and liver are very convinced, providing new findings regarding inflammation and muscle wasting n PDAC mouse. Did you find any presence of fibrosis in the liver and muscle? Such as TGFb, collagen?
  3. Could you please explain why you haven’t investigated protein expression of atrophy and apoptosis? Such as, western blotting for p62, caspase -3, caspase-9? Will be necessary to provide this data.
  4. Previous clinical translational work in cachectic patients demonstrated that muscle-stress response leads to muscle wasting through disruption of mitochondrial morphology, autophagy and apoptosis in gastric and colorectal cancer. A good overlap of these findings may provide better discussion at clinical levels. ( doi: 10.3390/cancers11091264 and https://doi.org/10.1016/j.clnu.2020.10.050)

Response:

Ad 1. Changed as suggested. Please note that we have now divided the former Figure 1 into two separate new Figures, i.e. new Figure 1 (gastrocnemius muscle) and new Figure 2 (soleus muscle), which both now include representative ATPase staining examples for a male and a female mouse of the WT and PDAC group, both with indications of fiber type in case of soleus muscle. Please see new Figure 1 (page 10) and new Figure 2 (page 11). We have added information in the Methods section, that  analysis in gastrocnemius muscle was limited to the larger homogenous (superficial) ‘white’ area consisting of type 2x/b only. 

Ad 2. This is an interesting suggestion, however, we actually have not (immuno)histochemically analyzed liver or muscle samples not stain for changes in this regard /TGFbeta, collagen), but we can state that we did not observe striking interstitial changes in muscle or signs of liver fibrosis. (Moreover, in line with reviewer’s suggestion, we are presently analyzing changes in hepatic amino acid, lipid, and glycogen handling, but these data are not yet available to date.)  

Ad. 3 This question is well-taken. Our present approach may be considered as somewhat comprehensive regarding histomorphology (inflammation, fiber size/composition, capillaries, neuromuscular junction; mitochondria) and included what can be considered a PCR- screening for further reported mechanisms of cachexia. The P62 or pro-/antiapoptotic signals on RNA level revealed no significant changes with PDAC or PanIN, however, we agree, this does not exclude changes on protein signaling levels and dedicated studies in this regard are warranted in the near future, evaluating different time windows of cachexia progression.

Ad. 4. Obviously, our study is focused on a relatively early stage of (pre)cachexia and can be considered as first ‘snap shots’ of PanIN and early PDAC stages. We agree, that various cachexia mechanisms, not presently reflected, may additionally be involved thereafter, though fiber atrophy is a major event also detected in a previous clinical cachexia study of our group (Ref. 71). We now added in the Discussion/ Limitations: the following sentences (page 28, par. 3, line 8; or total Discussion line number 179-186): ’’Notably, the present comparison of PDAC or PanIN stages to WT does not easily translate into clinical comparison between cachectic vs. non- or pre-cachectic cancer patients, which may represent different time windows of cachexia development through PDAC. As a more general remark, our study is unable to represent the complex clinical conditions of progressive cancer cachexia dealing with higher age, comorbidities, physical inactivity, pain or nutritional and psychosocial problems as imposed with various tumor entities, stages, treatments and complications.’’   

  • Reviewer

Open Review

English language and style

( ) Extensive editing of English language and style required
( ) Moderate English changes required
( ) English language and style are fine/minor spell check required
(x) I don't feel qualified to judge about the English language and style

Yes

Can be improved

Must be improved

Not applicable

Does the introduction provide sufficient background and include all relevant references?

(x)

( )

( )

( )

Is the research design appropriate?

( )

( )

(x)

( )

Are the methods adequately described?

( )

(x)

( )

( )

Are the results clearly presented?

( )

( )

(x)

( )

Are the conclusions supported by the results?

( )

(x)

( )

( )

Comments and Suggestions for Authors

Thank you for the invitation to review “Inflammation and wasting of skeletal muscles with intraepethelial neoplasia and pancreatic cancer in Kras-p53-mutant mice – when does cachexia start ?”. This manuscript by Hildebrandt and colleagues identify local factors initiating muscle wasting, inflammation, fiber cross-sectional area, composition, amino acid metabolism, capillarization, the integrity of neuromuscular junctions and mitochondria in hindlimb muscle of LSL-KrasG12D/+;LSL-TrP53R172H/+;Pdx1-Cre mice with intraepithelial-neoplasia 1-3 and PDAC compared to wild type.

Major revision:

The authors evaluate the CSA of the gastrocnemius and soleus muscles. In the mouse model, the soleus muscle has a high percentage of slow fibers of type 1 and 2a, a very low percentage of fibers 2x and 2b (- Bloemberg D, Quadrilatero J. Rapid determination of myosin heavy chain expression in rat, mouse, and human skeletal muscle using multicolor immunofluorescence analysis. PLoS One. 2012;7(4):e35273. - D'Amico D, Marino Gammazza A, Macaluso F, Paladino L, Scalia F, Spinoso G, Dimauro I, Caporossi D, Cappello F, Di Felice V, Barone R. Sex-based differences after a single bout of exercise on PGC1α isoforms in skeletal muscle: A pilot study. FASEB J. 2021 Feb;35(2):e21328.). On the contrary, the gastrocnemius muscle is rich in fibers 2a (red gastrocnemius) and 2x - 2b (white gastrocnemius) (- Bloemberg D, Quadrilatero J. Rapid determination of myosin heavy chain expression in rat, mouse, and human skeletal muscle using multicolor immunofluorescence analysis. PLoS One. 2012;7(4):e35273). The authors in this work should limit themselves to studying fibers 1 and 2a of the soleus muscle, and the fibers 2a, 2x - 2b (separate) for the gastrocnemius muscle.

Response:

We thank the Reviewer for this comment as it points out a necessary clarification in our manuscript.

We are well aware of the mentioned variable fiber composition in ‘red’ and ‘white’ gastrocnemius muscle depending on the area under study. In fact, we presently strictly analysed ‘white’ gastrocnemius, i.e. its larger homogeneous superficial area consisting of type 2x/b fibers only. This allowed to analyze >100 fibers within an intact tissue rectangle, which in fact is mostly not possible in ‘red’ gastrocnemius where fiber composition including type 2a (or even type 1) depends a lot on the distance from the soleus muscle. This mostly excludes an unbiased choice of rectangle area with >100 fibers, and this is also true for the plantaris muscle (not analyzed as well). We agree that ATPase staining does not differentiate between type 2x and 2b fibers, the present study, however, intended to comprehensively evaluate a broad spectrum of potential histomorphological, metabolic, ultrastructural and gene expression changes with PanIN and PDAC.      

Finally, we agree that this point needs to be clarified and, thus, have added the following sentence (page 5, first paragraph, line 8-9): ’’Notably, in gastrocnemius muscle, the analysis was limited to its larger homogenous ‘white’ (i. e. superficial) area that consists of type 2x/b fibers only’’

We also agree that the type 2x fiber fraction is rather small, though quite variable (please see newly added histological images in Figure 2). We certainly do not overinterprete changes within 2x, but believe that a ‘complete picture’ of soleus muscle alterations should include all fiber fractions detected by ATPase-staining. In fact, small changes in the large factions of type 1 and/or type 2a fiber may result in considerably large percentage changes in the small 2x fiber fraction (% count or % cross-sectional area overall) that might be functionally relevant. Besides, it may be of interest for the reader to observe, how type 2x fibers ‘respond’ to PDAC or PanIN within two different muscles (i.e. gastrocnemius and soleus muscle).

Please note, that the new Figures 1 and 2 – which we have improved in the revised manuscript - now present mean fiber CSA data and new image examples of male and female WT vs. PDAC mice separately for the (white) gastrocnemius (Figure 1) and soleus muscle (Figure 2), This information had formerly been summarized in old Figure 1.     

Minor revision:

  • Correct title: intraepethelial in intraepithelial;
  • Table 1 is cited before Figure 1, so it must appear before Figure 1;
  • Line 335 modify Fig.2c with Fig.1c;

Response: Changed as suggested 

  • Results on CSA show a significant decrease in fibers of female mice in PDAC mice compared to WT mice. But no differences in body weight were observed between the two groups. How do the authors explain this result?

Response:

This point is well taken and highlights the well-known clinical problem that weight changes are no reliable marker of muscle wasting. As stated in the beginning of the Results section (page 8, first paragraph, last sentence) and Discussion/Limitations (page 28, par. 2, line 2-3, and page 28, par. 3, line 8-14), important factors like ascites, tumor mass, metastases, changes in fat mass among others may have confounded a close relation between weight loss and muscle fiber atrophy at the presently studied early stage. Larger studies are warranted to assess contribution of all these factors including sex and other factors like activity, nutrition etc. to weight changes.

Reviewer 2 Report

Dear Authors,

Cancer Cachexia is a complex metabolic syndrome commonly associated with presence of systemic inflammation. The presented article is addressing an important question whether muscular inflammation plays a role in muscle wasting. The subject is extremely important for early understanding of cachexia mechanisms. However, minor comments/revisions were made bellow to improve the quality of the present publication.

  1. In the figure 1 you provided information about reduction in skeletal muscle fiber CSA in PDAC mice compared with WT. Could you please provide morphological analysis showing representative images to support these findings?
  2. CD68 and COX2 results in muscle and liver are very convinced, providing new findings regarding inflammation and muscle wasting n PDAC mouse. Did you find any presence of fibrosis in the liver and muscle? Such as TGFb, collagen?
  3. Could you please explain why you haven’t investigated protein expression of atrophy and apoptosis? Such as, western blotting for p62, caspase -3, caspase-9? Will be necessary to provide this data.
  4. Previous clinical translational work in cachectic patients demonstrated that muscle-stress response leads to muscle wasting through disruption of mitochondrial morphology, autophagy and apoptosis in gastric and colorectal cancer. A good overlap of these findings may provide better discussion at clinical levels. ( doi: 10.3390/cancers11091264 and https://doi.org/10.1016/j.clnu.2020.10.050)

Author Response

(The authors gave the same response as above.)

Reviewer 3 Report

Thank you for the invitation to review “Inflammation and wasting of skeletal muscles with intraepethelial neoplasia and pancreatic cancer in Kras-p53-mutant mice – when does cachexia start ?”. This manuscript by Hildebrandt and colleagues identify local factors initiating muscle wasting, inflammation, fiber cross-sectional area, composition, amino acid metabolism, capillarization, the integrity of neuromuscular junctions and mitochondria in hindlimb muscle of LSL-KrasG12D/+;LSL-TrP53R172H/+;Pdx1-Cre mice with intraepithelial-neoplasia 1-3 and PDAC compared to wild type.

Major revision:

The authors evaluate the CSA of the gastrocnemius and soleus muscles. In the mouse model, the soleus muscle has a high percentage of slow fibers of type 1 and 2a, a very low percentage of fibers 2x and 2b (- Bloemberg D, Quadrilatero J. Rapid determination of myosin heavy chain expression in rat, mouse, and human skeletal muscle using multicolor immunofluorescence analysis. PLoS One. 2012;7(4):e35273. - D'Amico D, Marino Gammazza A, Macaluso F, Paladino L, Scalia F, Spinoso G, Dimauro I, Caporossi D, Cappello F, Di Felice V, Barone R. Sex-based differences after a single bout of exercise on PGC1α isoforms in skeletal muscle: A pilot study. FASEB J. 2021 Feb;35(2):e21328.). On the contrary, the gastrocnemius muscle is rich in fibers 2a (red gastrocnemius) and 2x - 2b (white gastrocnemius) (- Bloemberg D, Quadrilatero J. Rapid determination of myosin heavy chain expression in rat, mouse, and human skeletal muscle using multicolor immunofluorescence analysis. PLoS One. 2012;7(4):e35273). The authors in this work should limit themselves to studying fibers 1 and 2a of the soleus muscle, and the fibers 2a, 2x - 2b (separate) for the gastrocnemius muscle.

Minor revision:

  • Correct title: intraepethelial in intraepithelial;
  • Table 1 is cited before Figure 1, so it must appear before Figure 1;
  • Line 335 modify Fig.2c with Fig.1c;
  • Results on CSA show a significant decrease in fibers of female mice in PDAC mice compared to WT mice. But no differences in body weight were observed between the two groups. How do the authors explain this result?

Author Response

(The authors gave the same response as above.)

Round 2

Reviewer 1 Report

Hildebrandt et al., submitted the manuscript, “Inflammation and wasting of skeletal muscles in Kras-p53-mutant mice with intraepithelial neoplasia and pancreatic cancer – when does cachexia start?”, and provided the data to explain the mechanisms of cachexia which occurred in mice with pancreatic cancer.

  1. I have several comments for this manuscript as follow,

In whole manuscript, the grammar needs to be edited. The examples in abstract are “Significant decreases in fiber CSA occurred with PDAC, but not PanIN1-3, in gastrocnemius (type 2x: -20.0%) and soleus (type 2a: -21.0%, type 1: -14.2%) muscle fibers with accentuation in male soleus (type 2a: -24.8%, type 1: -17.4%) and female gastrocnemius muscle (-29.6%). Significantly higher densities of endomysial CD68+-and cyclooxygenase-2+ (COX2+)- cells were detected with PDAC and, surprisingly, also with PanIN1-3 in both muscles, notably, with parallel increases of hepatic CD68+- and COX2+ cells. Moreover, in gastrocnemius muscle, suppressor-of-cytokine-3 (SOCS3) expressions was upregulated >2.7-fold with PanIN1A-3 and PDAC and the intracellular pool of proteinogenic amino acids and glutathione significantly increased with PanIN1A-3 compared to WT. In conclusion, the onset of fiber atrophy coincides with PDAC and high-grade local (and hepatic) inflammatory infiltration, without compromised microcirculation, innervation or mitochondria. Surprisingly, muscular and hepatic inflammation as well as SOCS3 upregulation were detectable already with precancerous PanINs together with (proteolytic) increase in free amino acids and glutathione.”

Please check other parts in this manuscript.

  1. The description in introduction is too complicated. Please focus on the title-related background.

  1. Please explain the meaning of the first sentence of introduction (Muscle wasting, beside loss of body fat, is a central, largely therapy-resistant feature of cancer cachexia with devastating effects on cancer patients prognosis, therapy outcome, quality of life and performance including their pulmonary function).

  1. Please explain the meaning of this sentence of introduction (As a multiorgan-syndrome cachexia is considered to involve circulating, i.e. systemic inflammatory or other catabolic factors (of tumor or multiple host origins) including altered hormonal profile, all of which converge in triggering myofibrillar net-proteolysis, increased hepatic amino acid consumption, and resting fat and liver energy expenditure as well as anorexia among others).

  1. The sentence in introduction, “Briefly, inflammatory, prooxidative or other catabolic mediators (e.g. TNF, IL6, PIF, ZAG) and the more recently identified IL8 (14) or MyD88 (15) are suggested to be responsible for proteasomal or other proteolytic pathway activation and/or inhibited protein synthesis, apoptosis, or impaired regeneration that arise, e.g. from activation of NF-B, upregulation in E3-ligases or autophagic genes and variably involve inhibition of Insulin-Akt-, activation of myostatin- or STAT3-pathways, mitochondrial dysfunction and other factors (6, 9,16-20) in muscle-specific fashion (21).”, is too long to be understood what the author want to tell us.

  1. Please explain the meaning of this sentence in introduction. “…preliminary data (presently included) of our group showed significant increases of CD68+ macrophages and COX2+ cells in the liver, i.e., downstream of the pancreas, already with PanIN 1A-B and PanIN 2-3.” What is COX2+ cells and what is the downstream of the pancreas?

  1. These sentences need to be edited. “We therefore actually aimed to follow up the exact onset of local inflammatory triggers and manifestation of skeletal muscle wasting by use of a mouse model that recapitulates mutant-based carcinogenesis with stepwise development of PanIN and PDAC, i.e. the established pancreas-targeted (knock-in) syngeneic KRASG12D/+; P53R172H/+; Pdx1-Cre+/+ (KPC) mouse model of PDAC. Thereby, the 2x-transgenic Kras-activating mutant mice PDAC develops mostly tumor-free PanIN stages, while transition to PDAC is slow and rare (6%). In contrast, 3x-transgenic Kras-activating and p53-inactivating mutant develops PDAC at a high rate (97%) and mimics well its clinical profile of cachexia, metastases, hemorrhagic ascites, whereby unrealistic tumor mass or inoculation sites (as involved in other cancer cachexia models) are avoided.”

  1. …..including E3-ligase upregulation and hepatic inflammation in C57/BL7 mice” Do you mean C57BL/6 mice?

  1. These sentences need to be edited. “For visualization of COX2+ cells, 7 μm cryo-cross-sections were processed as described above for CD68+ cells using polyclonal COX2 primary antibodies (rat anti-mouse, 1:50; Abcam, plc, Cambridge, UK) detected by HRP-conjugated antibodies (goat anti-rat, 1:100; AbD Serotec) incubated with DAB solution.

  1. In the materials and methods, “To assess the density of NMJ in quadriceps muscle, every 7th cross-section was treated for 10 min with 3% H2O2 to block endogenous peroxidases…”, please explain how to count every 7th cross-section.

  1. These sentences need to be edited. “For RT-qPCR, samples of gastrocnemius and soleus muscle samples were processed for qRT-PCR as recently described (42) using peqGOLD Isolation Systems TriFast (PEQLAB Biotechnologie GmbH, Erlangen, Germany) for RNA extraction according to the manufacturers protocol.

  1. The sentence in Results Part needs to be edited. “PanIN 1A-B was predominantly, but not exclusively present with 2x-trangenic (Kras-PDX) mutants, PanIN 2-3 was found with both, 2x-transgenic and 3x-trangenic (Kras-p53-PDX) mutants, and invasive PDAC, expectedly, occurred in 3x-transgenic mice only.

  1. Please explain the meaning of this sentence in Result. Age effect is a critical issue or not. “When controlled for age, which was significantly higher in PanIN or PDAC groups than in WT mice, however, also PDAC beside sex had a significant decreasing effect, indicating occurrence of cachexia at unknown tumor stage or mass, fat mass or ascites among other factors.

  1. Why analyze the fiber size in gastrocnemius and soleus muscle, but the fiber composition in soleus muscle only?

  1. These sentences in Results Part need to be edited. “ In soleus muscle, two-factorial ANOVA revealed that both, the factors PDAC and the factor sex (females vs. males), indeed significantly and independently decreased CSA of fiber type 1 (Figure 2a; PDAC: p=0.041, sex: p=0.017) and 2a (Figure 2b; PDAC: p=0.019, sex: p=0.007), while CSA of type 2x was significantly lower in females than in male mice (p=0.021) but not with PDAC compared to WT mice (p=0.946). When additionally controlled for age, PDAC and sex (female vs. male) still significantly decreased the CSA of type 1 fibers (PDAC: p=0.005, sex: p=0.012, age: p=0.042) and type 2a fibers (PDAC: p=0.044, sex: p=0.010, age: p=0.565) in soleus muscle. This approach additionally detected a significant decreasing effect of PDAC (p=0.035) beside sex (females vs. males, p=0.006) on type 2x fibers with no effect of age itself (p=0.082).

  1. Female mice have lower body weight than male mice. This is not necessary to compare in result.

  1. The sample group is too complicated. The mice with PanIN 1A-B and PanIN 2-3 were collected from 2x transgenic and 3x transgenic mouse models. The collected data can not be compared together.

  1. The sentence in Results Part needs to be edited. “The absence of any effect of all PanIN stages 1A-3 or only higher grade PanIN stage 2-3 on gastrocnemius or soleus muscle fibers of all types was further substantiated by ANOVA controlling for sex with or without consideration of age (analogue to testing for PDAC effects, as described above): This approach yielded no significant decreasing effect of PanIN stages 1A-3 or PanIN 2-3 alone on CSA of any fiber type in gastrocnemius or soleus muscle, while a significant decreasing effect of sex (females vs. males) was detectable irrespective of control for age on the CSA.

  1. The sentence in Results Part needs to be edited. Smaller fiber CSA can not indicate the muscle wasting directly. “Thus, as a main finding, PDAC, importantly however, not tumor-free PanIN stages (in 2x- or 3xtransgenic mice), led to muscle wasting in terms of significantly reduced mean fiber CSA compared to WT. Expectedly, thereby, female vs. male sex proved significantly and independently decreased fiber CSA, while age has little impact within the presently analyzed age span.

  1. As authors described, “Regarding changes in soleus muscle fiber composition (Figure 3a-c), a significant increase was observed in the rather small fraction of type 2x fibers with PDAC and PanIN 1A-B (Figure 3c) and associated with decreases in type 2a fiber fraction with PDAC, with significant differences in male mice only (Figure 3b).” What the critical point in these changes of different fiber types in different models?

  1. As authors described, “This was also true for calculated area fiber CSA per capillary contact as an inverse marker of capillary supply.” What the authors want to indicate?

  1. The sentence in Results Part needs to be edited. “Density of CD68+ macrophages (count per muscle fiber count) in gastrocnemius muscle (Figure 4a and b) was found to be significantly and progressively increased 2.8-fold, 4.0-fold, and 5.5-fold, with PanIN 1A-B, PanIN 2-3, and PDAC, respectively, when compared to WT, whereby the proinflammatory impact of PanIN 1A-3 overall or of PDAC was significant with (p=0.007 or p<0.001, respectively) and without (p=0.006 or p<0.001) control for impact of sex by ANOVA.

  1. The sentence in Results Part needs to be edited. “Linear regression analyses, including age (weeks) and pancreatic (histological) phenotype (defined as WT = 0, PanIN 1A-B = 1, PanIN2-3 = 2 and PDAC = 3) revealed a significant age-independent effect of pancreatic phenotype on gastrocnemius (p<0.001, age: p=0.229) and on soleus muscle (p<0.000, age: p=0.217) CD68+ macrophage density, with no additional impact of sex (not shown).” Please clearly describe the p-value indicated.

  1. Why and What is the conclusion in which COX2+ cells were increased in mice with PanIN 1A-B and PDAC but not PanIN 2-3?

  1. Please explain why compare the CD68+ macrophage and COX2+ cell density in skeletal muscle and liver? Have any reports indicated that increased CD68+ macrophages and COX2+ cells in liver are correlated with muscle wasting and cachexia?

  1. The sentence in Results Part needs to be edited. “Among the E3 ligases as atrophy markers in gastrocnemius muscle of PDAC mice, Atrogin-1, but not MuRF1 expression, showed an almost significant (p=0.065) 2.8-fold increase compared to WT, when controlled for the impact of sex, and a smaller trend (2.1-fold) towards increase with PanIN 2-3 (Table 3a). In soleus muscle (Table 3b), PDAC was associated with a 1.7-fold increase (p=0.056) in MuRF1 (significantly interacting with sex by ANOVA), but not in Atrogin-1 expression. Moreover, in gastrocnemius muscle, neither PDAC nor PanIN stages significantly changed the expression of Casp3, BAX or BCL2 (Table 3a) when compared to WT, while sex significantly affected Casp3 and BAX.

  1. The results in “gastrocnemius and soleus muscle” part let me confuse. That is not easy to understand what the hypothesis authors want to prove.

  1. These sentences in Results Part need to be edited. “Table 4 also presents a proteinogenic amino acids subgroup with protein-anabolic signaling potential, of which leucine, valine, proline and alanine, and, in part, glutamine and serine, showed significant increases in intramyocellular levels with PanIN 1A-B and/or PanIN 2-3 stages as well as with both combined PanIN stages controlled for the impact of sex by ANOVA, when compared to WT. Additionally, there were considerably smaller increases in all amino acid levels from WT levels with PDAC, being significantly different only in case of valine and, when controlled for sex, in case of leucine and glutamine.

  1. The sentence in Results Part needs to be edited. “As presented in Table 5, intramyocellular total and reduced GSH in gastrocnemius muscle were found to be significantly 1.2-fold increased in mice with both PanIN stages, but not with PDAC, which showed a significant decrease compared to PanIN 2-3, approaching control values of WT.

  1. Please explain what the subtitle indicated. “Correlation of SOCS3 and cytokines to inflammatory cells and of GSH redox state to SOCS3 in gastrocnemius muscle

  1. What is a significant correlation in this sentence? “A significant correlation to CD68 expression was also found for Il-1β (r=0.447 p<0.01), Il-6 (r=0.644 p<0.001) and Atrogin-1 (r=0.452 p=0.001) expressions (data not shown).

  1. Please explain this paragraph. “SOCS3 expression was significantly positively related to CD68 expression (r=0.352 p<0.05), however, not to CD68+ macrophage infiltration in gastrocnemius muscle (data not shown). A significant correlation to CD68 expression was also found for Il-1β (r=0.447 p<0.01), Il-6 (r=0.644 p<0.001) and Atrogin-1 (r=0.452 p=0.001) expressions (data not shown). Furthermore, SOCS3 expression was significantly related to COX2+ cell density in gastrocnemius muscle (r=0.394 p<0.05) (Figure 8a), but not to local COX2 expression. Among the cytokines under study, IL-6 was significantly related to COX2 expression (r=0.377 p<0.05) but not COX2+ cell density (data not shown).” What did authors mean that the different changes of cell density and gene expression? What points the authors want to indicate?

Author Response

Manuscript     cells-1653653     Revision 2

Hildebrandt, Keck et al.  ’’Inflammation and wasting of skeletal muscles with intraepithelial neoplasia and pancreatic cancer in Kras-p53-mutant mice – when does cachexia start ?’’

Point-to-Point Response to the Reviewer’s comments

Dear Editor,

we would like to thank the Reviewers again for their valuable time and the helpful comments, which were carefully considered for improving this manuscript. Please find below our Point-to-Point response to each of comments of Reviewer 1.

We have carefully revised all commented text sections and edited the whole manuscript with the help of an English native speaker.

All changes of this 2nd Revision 2 are marked up by green highlighting within the text body.

The yellow highlightings of the 1st Revision were kept at present.

We hope that our revised manuscript is now adequate for publication in CELLS.

With kind regards,

Sincerely

Wulf Hildebrandt (corresponding author)

Reviewer 1

Open Review

English language and style

(x) Extensive editing of English language and style required
( ) Moderate English changes required
( ) English language and style are fine/minor spell check required
( ) I don't feel qualified to judge about the English language and style

Yes

Can be improved

Must be improved

Not applicable

Does the introduction provide sufficient background and include all relevant references?

( )

( )

( )

(x)

Is the research design appropriate?

( )

( )

( )

(x)

Are the methods adequately described?

( )

( )

(x)

( )

Are the results clearly presented?

( )

( )

( )

(x)

Are the conclusions supported by the results?

( )

( )

(x)

( )

Comments and Suggestions for Authors

Hildebrandt et al., submitted the manuscript, “Inflammation and wasting of skeletal muscles in Kras-p53-mutant mice with intraepithelial neoplasia and pancreatic cancer – when does cachexia start?”, and provided the data to explain the mechanisms of cachexia which occurred in mice with pancreatic cancer.

  1. I have several comments for this manuscript as follow,

In whole manuscript, the grammar needs to be edited. The examples in abstract are “Significant decreases in fiber CSA occurred with PDAC, but not PanIN1-3, in gastrocnemius (type 2x: -20.0%) and soleus (type 2a: -21.0%, type 1: -14.2%) muscle fibers with accentuation in male soleus (type 2a: -24.8%, type 1: -17.4%) and female gastrocnemius muscle (-29.6%). Significantly higher densities of endomysial CD68+-and cyclooxygenase-2+ (COX2+)- cells were detected with PDAC and, surprisingly, also with PanIN1-3 in both muscles, notably, with parallel increases of hepatic CD68+- and COX2+ cells. Moreover, in gastrocnemius muscle, suppressor-of-cytokine-3 (SOCS3) expressions was upregulated >2.7-fold with PanIN1A-3 and PDAC and the intracellular pool of proteinogenic amino acids and glutathione significantly increased with PanIN1A-3 compared to WT. In conclusion, the onset of fiber atrophy coincides with PDAC and high-grade local (and hepatic) inflammatory infiltration, without compromised microcirculation, innervation or mitochondria. Surprisingly, muscular and hepatic inflammation as well as SOCS3 upregulation were detectable ‘already’ with precancerous PanINs together with (proteolytic) increase in free amino acids and glutathione.” Please check other parts in this manuscript.

  1. Response: We have edited language /grammar of the whole manuscript (see below). The Abstract section was improved as follows:

Significant decreases in fiber CSA occurred with PDAC, but not with PanIN1-3, as compared to WT: These were found in gastrocnemius (type 2x: -20.0%) and soleus (type 2a: -21.0%, type 1: -14.2%) muscle with accentuation in male soleus (type 2a: -24.8%, type 1: -17.4%) and female gastrocnemius muscle (-29.6%). Significantly higher densities of endomysial CD68+- and cyclooxygenase-2+ (COX2+)- cells were detected in mice with PDAC compared to WT. Surprisingly, CD68+- and COX2+-cell densities were also higher in mice with PanIN1A-3 in both muscles. Significant positive correlations existed between muscular and hepatic CD68+- or COX2+ cell densities. Moreover, in gastrocnemius muscle, suppressor-of-cytokine-3 (SOCS3) expressions was upregulated >2.7-fold with PanIN1A-3 and PDAC. The intracellular pools of proteinogenic amino acids and glutathione significantly increased with PanIN1A-3 compared to WT. Capillarization, NMJ, and mitochondrial ultrastructure remained unchanged with PanIN or PDAC. In conclusion, the onset of fiber atrophy coincides with the manifestation of PDAC and high-grade local (and hepatic) inflammatory infiltration without compromised microcirculation, innervation or mitochondria. Surprisingly, muscular and hepatic inflammation, SOCS3 upregulation and (proteolytic) increases in free amino acids and glutathione ‘already’ were detectable in mice with precancerous PanINs.

  1. The description in introduction is too complicated. Please focus on the title-related background.
  1. Response: We have revised, improved and simplified the Introduction avoiding too complicated sentences. We believe that readers not familiar with the multifactorial syndrome of cancer cachexia may profit from a short introduction including important review citations (Ref. Nr. 1-5, 9-12, 16-18). We actually do not focus on inflammation and oxidative stress only, but presently exclude important possible factors of cachexia such as compromised innervation, mitochondria or capillaries.

Briefly, the Introduction structure is as follows: 

In the 1st paragraph, we address clinical significance and prevalence (par 1, line 1-7), inflammatory mediators (par1, line 8-13), relevant catabolic pathways (par 1, line 13-16) and important contributions of liver and fat (par 1, line 16-19).

The 2nd paragraph addresses important gaps of knowledge regarding inflammation and highlights the potential role for SOCS3 (which we found to be upregulated in mice with precancerous PanINs and PDAC).

The 3rd paragraph points out, why the exact onset of cachexia needs to be studied (for the first time) in view of the inflammatory signals involved in precancerous PanIN stages and PDAC. The 4th paragraph, we explain why the 2x- 3x-transgenic mouse model of PDAC is adequate to this novel research question.

The 5th paragraph of the Introduction lists our measurements which comprehensively cover several presently discussed cachexia factors.

  1. Please explain the meaning of the first sentence of introduction (Muscle wasting, beside loss of body fat, is a central, largely therapy-resistant feature of cancer cachexia with devastating effects on cancer patients’ prognosis, therapy outcome, quality of life and performance including their pulmonary function).
  1. Response: We have revised this sentence as follows:

Cancer cachexia has been recognized as a complex, largely therapy-resistant syndrome that involves progressive muscle wasting and massively impairs the patients’ prognosis, therapy outcome, quality of life, exercise capacity and pulmonary function (1-6).

  1. Please explain the meaning of this sentence of introduction

 (As a multiorgan-syndrome cachexia is considered to involve circulating, i.e. systemic inflammatory or other catabolic factors (of tumor or multiple host origins) including altered hormonal profile, all of which converge in triggering myofibrillar net-proteolysis, increased hepatic amino acid consumption, and resting fat and liver energy expenditure as well as anorexia among others).

  1. Response: We have reorganized, simplified and clarified this section as follows:

Cancer-related muscle wasting has been attributed to circulating, i.e. systemic inflammatory, prooxidative or other catabolic factors of tumor or multiple host origins (9-13) such as TNF, IL6, PIF, ZAG or the more recently identified IL8 (14) or MyD88 (15). These factors may variably trigger muscular proteasomal or other proteolytic pathway activation, inhibit protein synthesis and muscle fiber regeneration or cause apoptosis. This may involve activation of the NF-kB-, STAT3- or myostatin-related pathways, inhibition of the insulin-Akt pathway, upregulation of E3-ligases or autophagic genes, mitochondrial dysfunction or other mechanisms (6,9,16-21). Besides, inflammation-driven increases in hepatic amino acid consumption and nitrogen losses as well as high resting fat and liver energy expenditure contribute to cachexia as a catabolic multi-organ syndrome (11-13).

  1. The sentence in introduction,

Briefly, inflammatory, prooxidative or other catabolic mediators (e.g. TNF, IL6, PIF, ZAG) and the more recently identified IL8 (14) or MyD88 (15) are suggested to be responsible for proteasomal or other proteolytic pathway activation and/or inhibited protein synthesis, apoptosis, or impaired regeneration that arise, e.g. from activation of NF-kB, upregulation in E3-ligases or autophagic genes and variably involve inhibition of Insulin-Akt-, activation of myostatin- or STAT3-pathways, mitochondrial dysfunction and other factors (6, 9,16-20) in muscle-specific fashion (21).”, is too long to be understood what the author want to tell us.

  1. Response: Please see our Response to 4. (We have revised/ improved the whole paragraph)
  1. Please explain the meaning of this sentence in introduction. “…preliminary data (presently included) of our group showed significant increases of CD68+ macrophages and COX2+ cells in the liver, i.e., downstream of the pancreas, already with PanIN 1A-B and PanIN 2-3.” What is COX2+ cells and what is the downstream of the pancreas?
  1. Response: We have clarified the meaning of this sentence as follows:

Of interest, in a preliminary study, we found significant increases of CD68+ macrophages and COX2+ cells in the liver (which receives portal venous blood from the pancreas) of mice with precursor lesions PanIN 1A-B and PanIN 2-3. 

  1. These sentences need to be edited. “We therefore actually aimed to follow up the exact onset of local inflammatory triggers and manifestation of skeletal muscle wasting by use of a mouse model that recapitulates mutant-based carcinogenesis with stepwise development of PanIN and PDAC, i.e. the established pancreas-targeted (knock-in) syngeneic KRASG12D/+; P53R172H/+; Pdx1-Cre+/+ (KPC) mouse model of PDAC. Thereby, the 2x-transgenic Kras-activating mutant mice PDAC develops mostly tumor-free PanIN stages, while transition to PDAC is slow and rare (6%). In contrast, 3x-transgenic Kras-activating and p53-inactivating mutant develops PDAC at a high rate (97%) and mimics well its clinical profile of cachexia, metastases, hemorrhagic ascites, whereby unrealistic tumor mass or inoculation sites (as involved in other cancer cachexia models) are avoided.”
  1. Response: We have clarified and improved this section as follows:

Our goal was to investigate the exact onset of local inflammatory triggers and manifestation of skeletal muscle wasting by use of a mouse model. The established pancreas-targeted (knock-in) syngeneic KRASG12D/+; P53R172H/+; Pdx1-Cre+/+ (KPC) mouse model of PDAC (34,35) recapitulates mutant-based carcinogenesis with stepwise development of PanIN and PDAC. The 2x-transgenic Kras-activating mutant mice develop mostly tumor-free PanIN stages, while transition to PDAC is slow and rare (6%). In contrast, the 3x-transgenic Kras-activating and p53-inactivating mutant mice develop PDAC at a high rate (97%) and mimic well its clinical profile of cachexia, metastases, hemorrhagic ascites, whereby unrealistic tumor mass or inoculation sites (that are required for other cancer cachexia models) can be avoided.

  1. …..including E3-ligase upregulation and hepatic inflammation in C57/BL7 mice” Do you mean C57BL/6 mice?
  1. Response: Changed as suggested to C57BL/6.
  1. These sentences need to be edited. “For visualization of COX2+ cells, 7 μm cryo-cross-sections were processed as described above for CD68+ cells using polyclonal COX2 primary antibodies (rat anti-mouse, 1:50; Abcam, plc, Cambridge, UK) detected by HRP-conjugated antibodies (goat anti-rat, 1:100; AbD Serotec) incubated with DAB solution.
  1. Response: We agree and have improved this sentence as follows:

For visualization of COX2+ cells, 7 µm cryo-cross-sections were processed as described above for CD68+ cells and incubated with polyclonal COX2 primary antibodies (rat anti-mouse, 1:50; Abcam, plc, Cambridge, UK).  Detection of COX2+ cells was performed using HRP-conjugated antibodies (goat anti-rat, 1:100; AbD Serotec) and incubation with DAB solution.

  1. In the materials and methods, “To assess the density of NMJ in quadriceps muscle,

           every 7th cross-section was treated for 10 min with 3% H2O2 to block

           endogenous peroxidases…”, please explain how to count every 7th cross-section.

  1. Response: We have clarified this point as follows:

To assess the density of NMJ in quadriceps muscle, every 7th out of 70 serial cross-sections was treated for 10 min with 3% H2O2 to block endogenous peroxidases..

  1. These sentences need to be edited. “For RT-qPCR, samples of gastrocnemius and soleus muscle samples were processed for qRT-PCR as recently described (42) using peqGOLD Isolation Systems TriFast™ (PEQLAB Biotechnologie GmbH, Erlangen, Germany) for RNA extraction according to the manufacturer’s protocol.
  1. Response: We have revised this section as follows:

For RT-qPCR, samples of gastrocnemius and soleus muscle samples were processed as recently described (42). Briefly, peqGOLD Isolation Systems TriFast™ (PEQLAB Biotechnologie GmbH, Erlangen, Germany) was used for RNA extraction according to the manufacturer’s protocol. 

  1. The sentence in Results Part needs to be edited. “PanIN 1A-B was predominantly, but not exclusively present with 2x-trangenic (Kras-PDX) mutants, PanIN 2-3 was found with both, 2x-transgenic and 3x-trangenic (Kras-p53-PDX) mutants, and invasive PDAC, expectedly, occurred in 3x-transgenic mice only.
  1. Response: We have clarified and simplified this sentence as follows:

Mice presenting with PanIN 1A-B were predominantly, but not exclusively, 2x-trangenic (Kras-PDX) mice. PanIN 2-3 lesions were found in both, 2x-transgenic and 3x-trangenic (Kras-p53-PDX) mice, and invasive PDAC occurred in 3x-transgenic mice only, as expected.

  1. Please explain the meaning of this sentence in Result. Age effect is a critical issue or not. “When controlled for age, which was significantly higher in PanIN or PDAC groups than in WT mice, however, also PDAC beside sex had a significant decreasing effect, indicating occurrence of cachexia at unknown tumor stage or mass, fat mass or ascites among other factors.
  1. Response: We have clarified this ANOVA analysis and presented p values for each of

            the factors PDAC/ PanINs, sex, and age:

Because female mice had significantly lower body weight than male mice and age was significantly higher in mice with PDAC or PanINs compared to WT (Table 1), we analysed the effect of PDAC or PanINs on body weight with control for age and sex by ANOVA. This procedure detected a significant effect of PDAC (p=0.002) and of sex (p<0.001), but not of age (p=0.146) on body weight. Moreover, body weight was significantly impacted by PanIN stages (p=0.002) when controlled for sex (p<0.001) and age (p<0.001).

  1. Why analyze the fiber size in gastrocnemius and soleus muscle, but the fiber composition in soleus muscle only?
  1. Response: Fiber composition in soleus muscle was analyzed in a rectangle area containing >150 fibers with a homogenous distribution pattern of fiber types across the area. Unfortunately, this procedure cannot be reliably applied to the relatively small and variable, soleus-adjacent part of the gastrocnemius muscle, which contains type 2a or even type 1 fibers in addition to type 2x/b fibers. In this area, fiber composition changes rapidly within short distance from soleus muscle. This basically excludes an unbiased choice of an representative area for histomorphometric analysis.
  1. These sentences in Results Part need to be edited. “ In soleus muscle, two-factorial ANOVA revealed that both, the factors PDAC and the factor sex (females vs. males), indeed significantly and independently decreased CSA of fiber type 1 (Figure 2a; PDAC: p=0.041, sex: p=0.017) and 2a (Figure 2b; PDAC: p=0.019, sex: p=0.007), while CSA of type 2x was significantly lower in females than in male mice (p=0.021) but not with PDAC compared to WT mice (p=0.946). When additionally controlled for age, PDAC and sex (female vs. male) still significantly decreased the CSA of type 1 fibers (PDAC: p=0.005, sex: p=0.012, age: p=0.042) and type 2a fibers (PDAC: p=0.044, sex: p=0.010, age: p=0.565) in soleus muscle. This approach additionally detected a significant decreasing effect of PDAC (p=0.035) beside sex (females vs. males, p=0.006) on type 2x fibers with no effect of age itself (p=0.082).
  1. Response: We have revised /simplified this section as follows:

In soleus muscle, two-factorial ANOVA showed that PDAC, independently of the factor sex (females vs. males), indeed significantly decreased CSA of fiber type 1 (Figure 2a; PDAC: p=0.041, sex: p=0.017) and 2a (Figure 2b; PDAC: p=0.019, sex: p=0.007) compared to WT. However, PDAC did not significantly impact CSA of type 2x (p=0.946), which was significantly lower in females than in male mice (p=0.021). With additional control for the factor age, PDAC still significantly contributed to a decrease in CSA of type 1 fibers (PDAC: p=0.005, sex: p=0.012, age: p=0.042) and type 2a fibers (PDAC: p=0.044, sex: p=0.010, age: p=0.565) in soleus muscle. This approach also detected a significant CSA-decreasing effect of PDAC (p=0.035) on type 2x fibers, which was independent of the factor sex (p=0.006) and the factor age (p=0.082).

  1. Female mice have lower body weight than male mice. This is not necessary to compare in result.
  1. Response: We agree that the sex-related difference is of course expected, however, because we include the factor sex in our ANOVA for detection of PDAC or PanIN effects on body weight or fiber CSA, we would prefer to present complete statistical information including posthoc testing as described in the Methods statistic section.  
  1. The sample group is too complicated. The mice with PanIN 1A-B and PanIN 2-3 were collected from 2x transgenic and 3x transgenic mouse models. The collected data can not be compared together.
  1. Response: In view of the inflammatory background caused by the progression of PanIN stages, we consider it valuable to present mean data separately for the PanIN 1A-B, and PanIN 2-3 stages. As a main question, we presently investigate, whether cachexia-relevant changes occur before or after manifestation of PDAC and accordingly use ANOVA with the factor PanIN 2-3 or PanIN 1A-3 stages (see Methods statistic). We agree that the impact of genotype independent of PanINs, sex, age or other factors may warrant further analyses in a larger study population. Please note that this KPC mouse model involves the pancreas-specific Pdx1-promotor.
  1. The sentence in Results Part needs to be edited. “The absence of any effect of all PanIN stages 1A-3 or only higher grade PanIN stage 2-3 on gastrocnemius or soleus muscle fibers of all types was further substantiated by ANOVA controlling for sex with or without consideration of age (analogue to testing for PDAC effects, as described above): This approach yielded no significant decreasing effect of PanIN stages 1A-3 or PanIN 2-3 alone on CSA of any fiber type in gastrocnemius or soleus muscle, while a significant decreasing effect of sex (females vs. males) was detectable irrespective of control for age on the CSA.
  1. Response: This section summarizes, that no significant impact of PanIN 2-3 or of PanIN 1A-3 on gastrocnemius or soleus muscle CSA was detected by ANOVA with control for sex (and age). This procedure was described beforehand for the impact of PDAC. We now avoid partly redundant information on the contribution of gender and age. We have revised and simplified this section as follows:

Notably, no significant atrophy was detected in both PanIN stages (without PDAC) in any of these fiber types and muscles under study, except for gastrocnemius muscle fibers with PanIN 1A-B in females (Figure 1a). Also, when applying ANOVA with control for sex and age (as described above for testing the impact of PDAC), no significant impact of PanIN stages 1A-3 or of P anIN stage 2-3 was detected on the CSA of gastrocnemius fibers or any soleus muscle fiber type.

  1. The sentence in Results Part needs to be edited. Smaller fiber CSA can not indicate the muscle wasting directly. “Thus, as a main finding, PDAC, importantly however, not tumor-free PanIN stages (in 2x- or 3xtransgenic mice), led to muscle wasting in terms of significantly reduced mean fiber CSA compared to WT. Expectedly, thereby, female vs. male sex proved significantly and independently decreased fiber CSA, while age has little impact within the presently analyzed age span.
  1. Response: We agree and have changed the sentence accordingly:

Thus, as a main finding, PDAC, but not the tumor-free PanIN stages (in 2x- or 3xtransgenic mice), was associated with a significant reduction in mean fiber CSA compared to WT.

  1. As authors described, “Regarding changes in soleus muscle fiber composition (Figure 3a-c), a significant increase was observed in the rather small fraction of type 2x fibers with PDAC and PanIN 1A-B (Figure 3c) and associated with decreases in type 2a fiber fraction with PDAC, with significant differences in male mice only (Figure 3b).” What the critical point in these changes of different fiber types in different models?
  1. Response: We agree that this is a secondary outcome of the study and not further discussed in detail. However, rather small changes in the type 1 and/or 2a fraction may lead to large percentage changes in the type 2x fiber fraction. Therefore, we would prefer to present the ‘complete picture’ of alterations in soleus muscle. Please note, that this is the first description of altered muscle histomorphometry in this KPC mouse model.  
  1. As authors described, “This was also true for calculated area fiber CSA per capillary contact as an inverse marker of capillary supply.” What the authors want to indicate?
  1. Response: We have improved this sentence as follows:

Likewise, the ratio of fiber CSA /capillary contact, which was calculated as an inverse marker of capillary supply for each fiber type, did not significantly differ between mice with PDAC- or PanINs and WT.  

  1. The sentence in Results Part needs to be edited. “Density of CD68+ macrophages (count per muscle fiber count) in gastrocnemius muscle (Figure 4a and b) was found to be significantly and progressively increased 2.8-fold, 4.0-fold, and 5.5-fold, with PanIN 1A-B, PanIN 2-3, and PDAC, respectively, when compared to WT, whereby the proinflammatory impact of PanIN 1A-3 overall or of PDAC was significant with (p=0.007 or p<0.001, respectively) and without (p=0.006 or p<0.001) control for impact of sex by ANOVA.
  1. Response: We have revised and divided this sentence into two sentences:

The density of CD68+ macrophages (count per muscle fiber count) in gastrocnemius muscle (Figure 4a and b) was found to be significantly and progressively increased 2.8-fold, 4.0-fold, and 5.5-fold, with PanIN 1A-B, PanIN 2-3, and PDAC, respectively, as compared to WT. This proinflammatory impact of PanINs or PDAC was significant with (p=0.007 or p<0.001, respectively) and without (p=0.006 or p<0.001) control for the factor sex by ANOVA.

  1. The sentence in Results Part needs to be edited. “Linear regression analyses, including age (weeks) and pancreatic (histological) phenotype (defined as WT = 0, PanIN 1A-B = 1, PanIN2-3 = 2 and PDAC = 3) revealed a significant age-independent effect of pancreatic phenotype on gastrocnemius (p<0.001, age: p=0.229) and on soleus muscle (p<0.000, age: p=0.217) CD68+ macrophage density, with no additional impact of sex (not shown).” Please clearly describe the p-value indicated. 
  1. Response: We have revised this sentence and more clearly assigned the p value to the factor pancreatic phenotype within this regression procedure:

Linear regression analyses, including pancreatic (histological) phenotype (defined as WT = 0, PanIN 1A-B = 1, PanIN2-3 = 2 and PDAC = 3), age (weeks) and sex (males vs. females) revealed a significant, independent effect of the pancreatic phenotype on CD68+ macrophage density in gastrocnemius and soleus muscle (p<0.001 and p<0.000, respectively) with no significant impact of sex or age (not shown).

  1. Why and What is the conclusion in which COX2+ cells were increased in mice with PanIN 1A-B and PDAC but not PanIN 2-3?
  1. Response: This point is well-taken. We presently do not know, why increases in COX+ cell densities reach significance in PDAC and PanIN 1A-B (and PanINs overall) but not in PanIN 2-3. However, because this finding is consistent across both muscles, we consider it worth reporting. Notably, the knowledge on systemic inflammation with PanIN progression is very limited.
  1. Please explain why compare the CD68+ macrophage and COX2+ cell density in skeletal muscle and liver? Have any reports indicated that increased CD68+ macrophages and COX2+ cells in liver are correlated with muscle wasting and cachexia?
  1. Response: Available data on inflammatory cell infiltration in muscle or liver with PDAC are very limited and restricted to early stages of cachexia (Ref 6,23,24,78). As a novel finding, we presently report increased CD68+ and COX2+ cell densities not only with PDAC but also with precancerous PanINs. Our finding of a positive correlation (overall) between muscular and hepatic inflammatory cell densities is novel and points at a systemic role of hepatic inflammation that warrants further studies. Generally, the role of the liver in cancer cachexia appears understudied and may start early on (ref. 12, 78) or even before manifestation of PDAC, as suggested for the first time by our data.  
  1. The sentence in Results Part needs to be edited. “Among the E3 ligases as atrophy markers in gastrocnemius muscle of PDAC mice, Atrogin-1, but not MuRF1 expression, showed an almost significant (p=0.065) 2.8-fold increase compared to WT, when controlled for the impact of sex, and a smaller trend (2.1-fold) towards increase with PanIN 2-3 (Table 3a). In soleus muscle (Table 3b), PDAC was associated with a 1.7-fold increase (p=0.056) in MuRF1 (significantly interacting with sex by ANOVA), but not in Atrogin-1 expression. Moreover, in gastrocnemius muscle, neither PDAC nor PanIN stages significantly changed the expression of Casp3, BAX or BCL2 (Table 3a) when compared to WT, while sex significantly affected Casp3 and BAX.
  1. Response: We have improved and simplified this section to improve readability:

Among the E3 ligases that were used as atrophy markers in gastrocnemius muscle of PDAC mice, Atrogin-1, but not MuRF1 expression, showed an almost significant (p=0.065) 2.8-fold increase compared to WT, when controlled for the impact of sex. A smaller trend (2.1-fold) towards an Atrogin-1 increase was observed with PanIN 2-3 (Table 3a). In soleus muscle (Table 3b), PDAC was associated with a 1.7-fold increase (p=0.056) in MuRF1 (significantly interacting with sex by ANOVA), but not in Atrogin-1 expression. Moreover, in gastrocnemius muscle, neither PDAC nor PanIN stages significantly affected the expression of Casp3, BAX or BCL2 (Table 3a) when compared to WT, while gender significantly affected Casp3 and BAX. In soleus muscle, PDAC or sex showed no significant effect on the expression of Casp3, BAX or BCL2 (Table 3b), while p62 was significantly affected by sex only.

  1. The results in “gastrocnemius and soleus muscle” part let me confuse. That is not easy to understand what the hypothesis authors want to prove.
  1. Response: We understand that our rather comprehensive set of measurements may in part be seen as a hypothesis-generating screening for early precancerous and PDAC-related changes. However, our selection of transcripts was based on state-of-the-art review articles on cancer cachexia cited in the Introduction section. 
  1. These sentences in Results Part need to be edited. “Table 4 also presents a proteinogenic amino acids subgroup with protein-anabolic signaling potential, of which leucine, valine, proline and alanine, and, in part, glutamine and serine, showed significant increases in intramyocellular levels with PanIN 1A-B and/or PanIN 2-3 stages as well as with both combined PanIN stages controlled for the impact of sex by ANOVA, when compared to WT. Additionally, there were considerably smaller increases in all amino acid levels from WT levels with PDAC, being significantly different only in case of valine and, when controlled for sex, in case of leucine and glutamine.
  1. Response: We agree and have revised this section as follows:

Table 4 also presents a subgroup of proteinogenic amino acids with protein-anabolic signaling potential. Here leucine, valine, proline and alanine, and, in part, glutamine and serine, showed significant increases in intramyocellular levels with PanIN 1A-B and/or PanIN 2-3 stages as well as with both combined PanIN stages when compared to WT levels with control for the impact of sex by ANOVA. Additionally, there were considerably smaller increases in all amino acid levels with PDAC as compared to WT levels. Only in the case of valine and, when controlled for sex, in the case of leucine and glutamine were these differences significant.

  1. The sentence in Results Part needs to be edited. “As presented in Table 5, intramyocellular total and reduced GSH in gastrocnemius muscle were found to be significantly 1.2-fold increased in mice with both PanIN stages, but not with PDAC, which showed a significant decrease compared to PanIN 2-3, approaching control values of WT.
  1. Response: We have revised this section and divided it in two sentences as follows: 

As presented in Table 5, intramyocellular total and reduced GSH in gastrocnemius muscle were found to be significantly increased 1.2-fold in mice with PanIN stages, but not with PDAC. The latter showed a significant decrease compared to PanIN 2-3, approaching control values of WT

  1. Please explain what the subtitle indicated. “Correlation of SOCS3 and cytokines to inflammatory cells and of GSH redox state to SOCS3 in gastrocnemius muscle
  1. Response: We have revised and simplified this title as follows:

Correlations between inflammatory signals, SOCS3 expression, and the GSH redox state in gastrocnemius muscle

  1. What is a significant correlation in this sentence? “A significant correlation to CD68 expression was also found for Il-1β (r=0.447 p<0.01), Il-6 (r=0.644 p<0.001) and Atrogin-1 (r=0.452 p=0.001) expressions (data not shown).
  1. Response: We revised this sentence as follows:

Moreover, significant correlations existed between the expressions of Il-1β (r=0.447 p<0.01), Il-6 (r=0.644 p<0.001) or Atrogin-1 (r=0.452 p=0.001) and the expression of CD68 (data not shown).

  1. Please explain this paragraph. “SOCS3 expression was significantly positively related to CD68 expression (r=0.352 p<0.05), however, not to CD68+ macrophage infiltration in gastrocnemius muscle (data not shown). A significant correlation to CD68 expression was also found for Il-1β (r=0.447 p<0.01), Il-6 (r=0.644 p<0.001) and Atrogin-1 (r=0.452 p=0.001) expressions (data not shown). Furthermore, SOCS3 expression was significantly related to COX2+ cell density in gastrocnemius muscle (r=0.394 p<0.05) (Figure 8a), but not to local COX2 expression. Among the cytokines under study, IL-6 was significantly related to COX2 expression (r=0.377 p<0.05) but not COX2+ cell density (data not shown).” What did authors mean that the different changes of cell density and gene expression? What points the authors want to indicate?
  1. Response: Since little data on local muscular inflammation with cancer cachexia are published, we consider it valuable to report significant correlations between CD68+ or COX+ cell densities, relevant cytokine transcripts and SOCS3 expression. We have, however, reduced the information on non-significant correlations to clarify this section as follows:

SOCS3 expression was significantly positively related to CD68 expression (r=0.352 p<0.05), however, not to CD68+ macrophage infiltration in gastrocnemius muscle (data not shown). Moreover, significant correlations existed between the expressions of Il-1β (r=0.447 p<0.01), Il-6 (r=0.644 p<0.001) or Atrogin-1 (r=0.452 p=0.001) and the expression of CD68 (data not shown). Furthermore, SOCS3 expression was significantly related to COX2+ cell density in gastrocnemius muscle (r=0.394 p<0.05) (Figure 8a). Among the cytokines under study, IL-6 was found to be significantly related to COX2 expression (r=0.377 p<0.05) (data not shown).

Reviewer 3

Open Review

English language and style

( ) Extensive editing of English language and style required
( ) Moderate English changes required
( ) English language and style are fine/minor spell check required
(x) I don't feel qualified to judge about the English language and style

Yes

Can be improved

Must be improved

Not applicable

Does the introduction provide sufficient background and include all relevant references?

(x)

( )

( )

( )

Is the research design appropriate?

(x)

( )

( )

( )

Are the methods adequately described?

(x)

( )

( )

( )

Are the results clearly presented?

(x)

( )

( )

( )

Are the conclusions supported by the results?

(x)

( )

( )

( )

Comments and Suggestions for Authors

With these changes the paper has improved

Submission Date

10 March 2022

Date of this review

07 Apr 2022 07:33:46

Reviewer 3 Report

With these changes the paper has improved

Author Response

(The authors gave the same response as above.)

Round 3

Reviewer 1 Report

No further comments.